# Biased Generalization in Diffusion Models

Jérôme Garnier-Brun [* 1]   Luca Biggio [* 1]   Davide Beltrame [1]   Marc Mézard [1]   Luca Saglietti [1]

## Abstract

Generalization in generative modeling is defined as the ability to learn an underlying distribution from a finite dataset and produce novel samples, with evaluation largely driven by held-out performance and perceived sample quality. In practice, training is often stopped at the minimum of the test loss, taken as an operational indicator of generalization. We challenge this viewpoint by identifying a phase of *biased generalization* during training, in which the model continues to decrease the test loss while favoring samples with anomalously high proximity to training data. By training the same network on two disjoint datasets and comparing the mutual distances of generated samples and their similarity to training data, we introduce a quantitative measure of bias and demonstrate its presence on real images. We then study the mechanism of bias, using a controlled hierarchical data model where access to the exact denoiser and ground-truth statistics allows us to precisely characterize its onset. We attribute this phenomenon to the sequential nature of feature learning in deep networks, where coarse structure is learned early in a data-independent manner, while finer features are resolved later in a way that increasingly depends on individual training samples. Our results show that early stopping at the test loss minimum, while optimal under standard generalization criteria, may be insufficient for privacy-critical applications.

## 1. Introduction

Generative AI can now produce text, images, and videos with a level of realism that was hardly imaginable only a few years ago, an achievement that also introduces profound societal challenges (Weidinger et al., 2023). Whatever the medium, two questions stand at the center of current research: (i) does the generated content possess sufficient *quality* to appear authentic, and (ii) is it truly *novel* rather than a near-duplicate or patchwork of examples from the training set? More precisely, considering the task of learning to generate from a target distribution $P_0 : \mathbb{R}^d \to \mathbb{R}$ given $n$ fair samples $\{x^\mu\}_{\mu=1,\ldots,n}$, one should ensure that the generative process (i) samples according to a distribution $\tilde{P}_0^\theta$ that has a small distance to $P_0$, i.e., appears to achieve genuine *generalization*; and (ii) does not generate individual samples $x$ that are anomalously close to some training point $x^\mu$, displaying some form of *memorization*.

The interplay between generalization and memorization is particularly relevant in the context of generative diffusion (Sohl-Dickstein et al., 2015). There, as neural networks are typically trained to denoise a finite number of training samples, the minimum training loss is necessarily achieved by memorizing training examples (Gu et al., 2025). The prevailing view is that diffusion models generalize when they fail to memorize (Yoon et al., 2023), whether due to limited architectural capacity (George et al., 2026) or favorable inductive biases (Kadkhodaie et al., 2024; Kamb & Ganguli, 2025; Niedoba et al., 2025). Recent work extends this picture to the training dynamics: diffusion models generalize on the way to memorization, with a plateau of good generative performance preceding the onset of overfitting (Bonnaire et al., 2025; Favero et al., 2025b); see also Montanari & Urbani (2026) in a different context. This motivates early stopping at the minimum of the test loss, mirroring standard practice in supervised learning.

Nonetheless, as demonstrated by Carlini et al. (2023), even models that appear to generalize well and are reported not to overfit in training, such as Imagen (Saharia et al., 2022), sometimes reproduce training samples almost exactly. This apparent paradox suggests that the dichotomy between generalization and memorization may be too coarse.

The *denoising score-matching* (DSM) loss used to train diffusion models can be shown to be an upper bound of the Kullback-Leibler (KL) divergence between the true data distribution $P_0$ and the trained model distribution $\tilde{P}_0^\theta$ (Song et al., 2021). However, minimizing this loss does not preclude biased generation towards training examples. This disconnect, which was already identified by van den Burg & Williams (2021) in the context of variational autoencoders,

---
[*]Equal contribution  [1]Department of Computing Sciences, Bocconi University, Milan, Italy. Correspondence to: Jérôme Garnier-Brun <jerome.garnier@unibocconi.it>.

*Proceedings of the 43rd International Conference on Machine Learning*, Seoul, South Korea. PMLR 306, 2026. Copyright 2026 by the author(s).

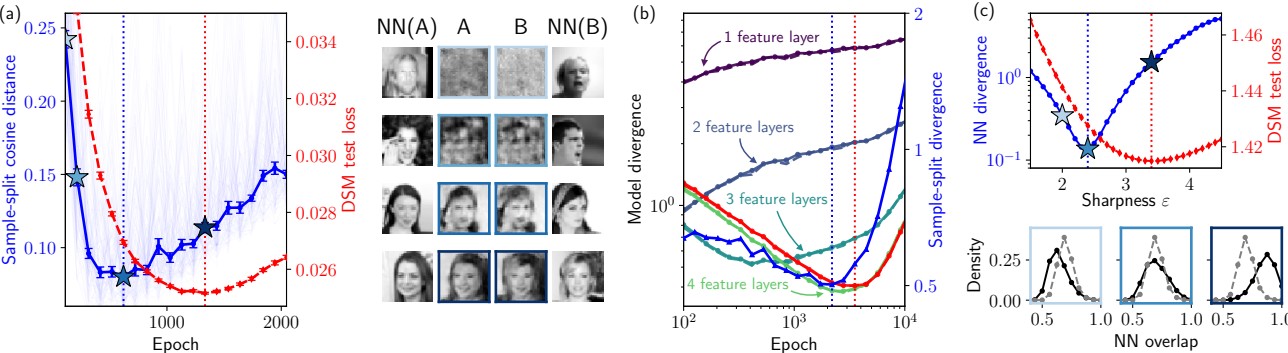

*Figure 1.* Biased generalization emerges before overfitting across models and settings. (a) Sample-split analysis on CelebA: we compare two denoising diffusion models trained on non-overlapping data slices. Left: cosine distance between generated samples (left axis) and denoising score matching (DSM) test loss (right axis) during training, means over 15 models with standard errors. Generated samples become maximally similar before the test loss is minimal, indicating the onset of *biased generalization* while the test loss is still decreasing. Colored stars mark epochs selected for visualization. Right: samples generated at the starred epochs by a model trained on each of the database split A/B (central columns), with nearest neighbors (NN) from each training split (side columns). Early in training, both models evolve similarly and sample quality improves; near the test-loss minimum, generated samples can differ substantially across splits and may get close to training examples, showing a bias without exact memorization. (b) Neural network trained on a controlled hierarchical dataset. Model-oracle divergence (left axis), representing the distance of the learned denoiser to the exact denoiser (red) and four coarser versions of the latter that account for lower-level features (as indicated by arrows). Sample-split analysis (right axis, blue) measuring the distance between denoising posterior mean of models trained on disjoint datasets. All posterior means are computed on test samples (w.r.t the model's training data), noised up to a critical diffusion time $t/T = 0.15$. The biased generalization phase is seen between the minimum of the sample-split curve (blue) and the minimum of the model-exact oracle divergence (red). It takes place when the models start resolving finer-scale features (light green). (c) Training-free diffusion model on the same hierarchical data, parametrized by a sharpness parameter $\varepsilon$ controlling the concentration of probability mass around the training data. Top: divergence between generated data and ground truth distributions of distances to the nearest neighbor (NN) in the training set (left axis) and DSM test loss (right axis) as a function of the sharpness parameter $\varepsilon$, showing sizeable bias at the test-loss minimum. Stars indicate selected values of $\varepsilon$. Bottom: in-training NN overlap (1 - normalized distance) distributions at selected sharpness levels, comparing model samples (solid black) to the ground truth (dashed gray).

highlights the need to move beyond aggregate generalization metrics and examine more localized signs of memorization or bias towards training-data in generative diffusion. Understanding the crossover between training-data-independent generalization and inevitable overfitting is essential to ensure that trained models are sufficiently accurate while not violating privacy or copyright-related constraints in relation to the training data.

In this paper, we tackle this issue in denoising diffusion probabilistic models (DDPM) (Ho et al., 2020). Prior efforts to understand the transition from generalization to memorization have focused on analytically tractable but simplified models (Li et al., 2023; George et al., 2026), or on empirical studies using real images (Gu et al., 2025; Ross et al., 2025), the more detailed understanding being obtained by a mixture of these two strategies (Bonnaire et al., 2025), see also (Favero et al., 2025b). Here, we study biased generalization using a similar two-pronged approach. We first demonstrate the emergence of bias before overfitting in models trained on real images. We then study the phenomenon in a controlled hierarchical data model with tunable long-range correlations, allowing us to precisely characterize the onset of bias thanks to the availability of ground-truth quantities.

**Contributions.** The key takeaway of our paper is: *In diffusion models, generalization and memorization can coexist, behaving as orthogonal—rather than opposite—axes of generative behavior.* More precisely:

- We show that diffusion models trained on real images (CelebA) exhibit a phase of *biased generalization*: two models trained on disjoint datasets start producing increasingly different outputs before either shows signs of overfitting, as measured by an increasing test loss, reflecting a growing bias toward their respective training samples (Fig. 1(a)).

- We refine these findings in a controlled hierarchical data model, where trained denoisers can be analyzed with access to the exact posterior mean and ground-truth statistics, allowing us to confirm the emergence of bias through multiple complementary diagnostics (Fig. 1(b)).

- We provide a mechanistic account of why bias can emerge before overfitting, connecting the phenomenon to the sequential nature of feature learning in deep networks: while coarse structure is learned in a data-independent manner, the extraction of finer features becomes increasingly reliant on the available samples, yet generalization can continue to improve (Fig. 1(b)).

- We illustrate the early onset of bias in a training-free

simple diffusion model on our hierarchical data, showing that *biased generalization* is not an artifact of the inductive bias of neural networks or of SGD-based optimization dynamics (Fig. 1(c)).

Section 2 introduces the diffusion framework and the metrics used to detect bias, while the results are presented starting from Sec. 3. We refer the reader to Appendix A for an extended discussion of the positioning of our paper with respect to related works.

## 2. Diffusion models and bias

### 2.1. Generative diffusion framework

Although the phenomenon of biased generalization is more general, we focus on DDPM (Ho et al., 2020). In this framework, the forward process gradually corrupts data by adding Gaussian noise: at timestep $0 \leq t \leq T$, a clean sample $\boldsymbol{x}_0$ becomes

$$\boldsymbol{x}_t = \sqrt{\overline{\alpha}_t}\boldsymbol{x}_0 + \sqrt{1 - \overline{\alpha}_t}\boldsymbol{\xi}_t, \qquad \boldsymbol{\xi}_t \sim \mathcal{N}(0, \mathbf{I}_d) \quad (1)$$

where $\overline{\alpha}_t$ controls the noise level, monotonically decreasing from $\overline{\alpha}_0 = 1$ to $\overline{\alpha}_T \approx 0$. Generation proceeds by reversing this process, which requires estimating the *posterior mean*

$$\hat{\boldsymbol{x}}_0(\boldsymbol{x}_t) = \mathbb{E}[\boldsymbol{x}_0 \mid \boldsymbol{x}_t], \quad (2)$$

at each noise level. A neural network is trained to minimize the DSM loss on corrupted training samples, leading to an approximate posterior mean $\hat{\boldsymbol{x}}_0^\theta$ (or equivalently an estimate of the added noise $\hat{\boldsymbol{\xi}}_t^\theta$).

**Generalization and memorization.** During training, supervision is local (restricted to noisy neighborhoods of the training samples in input space) yet most generative trajectories spend the majority of their evolution outside these regions (Song et al., 2025). The ability of the model to denoise unseen inputs, measured by the test loss, thus reflects underlying distributional structure and implies generalization. This notion, however, is in tension with the training objective. When training on a finite dataset $\{\boldsymbol{x}_0^\mu\}_{\mu=1,\dots,n}$, the denoising score-matching training loss is minimized by the *empirical* denoiser, corresponding to the posterior mean computed with respect to the empirical data distribution concentrated on the training samples. As noted by Gu et al. (2025), a sufficiently expressive network perfectly minimizing the training loss will therefore eventually memorize the training set. Early stopping, i.e. halting training at the minimum of the test loss, is commonly invoked to mitigate this tension, see e.g. (Li et al., 2023), as it can formally be shown that the test DSM loss is an upper bound to the $D_{\mathrm{KL}}(P_0 \| \tilde{P}_0^\theta)$ (Song et al., 2021).

**Detecting memorization.** A common approach for detecting memorization is to compare generated samples to

their nearest neighbors in the training set. For instance, on real datasets, Yoon et al. (2023) define a generated sample $\boldsymbol{x}$ as *replicating* a training point if

$$\frac{\|\boldsymbol{x} - \boldsymbol{x}_0^a\|_2}{\|\boldsymbol{x} - \boldsymbol{x}_0^b\|_2} < \frac{1}{3}, \quad (3)$$

where $\boldsymbol{x}_0^a$ and $\boldsymbol{x}_0^b$ denote the closest and second-closest samples in the training dataset under the $\ell_2$ norm. Empirically, such proximity-based indicators are found to become non-negligible only after the test loss starts increasing, i.e., in the overfitting regime. This observation supports the widespread interpretation of memorization as a failure mode that emerges in opposition to generalization. In high-dimensional settings, these two generative regimes have been found to occur on different time scales (Bonnaire et al., 2025; Favero et al., 2025b) separated by a long plateau of good generalizing behavior. By contrast, we find that on structured finite-dimensional data, the generalization phase can itself be subdivided into a universal—sample independent—and a biased phase.

### 2.2. Measuring bias in practice

We define *bias* in a generative model as the emergence of anomalous similarity between generated samples and the training data, relative to what would be expected under i.i.d. sampling from the underlying data distribution. More formally, following the spirit of Carlini et al. (2023):

**Definition 2.1** (($d, \delta, \lambda$)-biased generative model)**.** Given a distance $d$, a radius $\delta$, and a tolerance $\lambda$, a generative model $\hat{P}$ is said to be ($d, \delta, \lambda$)-biased toward its training set $\{\boldsymbol{x}^\mu\}_{\mu=1,\dots,n}$ relative to the underlying data distribution $P$ if there exists $\mu$ such that

$$\mathbb{P}_{\boldsymbol{x} \sim \hat{P}}[d(\boldsymbol{x}, \boldsymbol{x}^\mu) < \delta] - \mathbb{P}_{\boldsymbol{x} \sim P}[d(\boldsymbol{x}, \boldsymbol{x}^\mu) < \delta] > \lambda. \quad (4)$$

An analogous definition can be formulated at the level of the learned denoising function, comparing its outputs to either the ground-truth posterior mean or to a reference denoiser trained on independent data. In both cases, the formal statement displaces the question onto the choice of ($d, \delta, \lambda$), which is inherently context-dependent: a differential-privacy standpoint requires arbitrarily small thresholds for any training point, whereas generative-fidelity considerations admit much coarser ones. Rather than fixing these parameters a priori, our aim is to characterize bias along training and identify the regime in which it emerges, using the diagnostics described below.

Unlike overt memorization, such bias can be subtle and may arise well before individual training samples are nearly reproduced. Nevertheless, this notion is relevant in settings where data privacy is critical, and the systematic reuse of training-specific features may be undesirable even in the absence of exact sample replication. Detecting bias therefore

requires a reference notion of unbiased behavior, against which model outputs can be compared. Bias can be probed through complementary observables, either at the level of generated samples or at the level of the learned denoising function.

**Sample-level.** Rather than relying on binary memorization criteria, we consider the full distribution of distances between generated samples and their nearest neighbors in the training set. Along training, the generated data should shift from being anomalously far (random initialization) to anomalously close (eventual memorization) to training samples, see Fig. 1(c). By analyzing the Kullback-Leibler divergence from the unbiased distribution—which we term the *nearest-neighbor divergence*—one can detect systematic deviations toward smaller distances, indicating biased generation toward the training data. For discrete-valued data with finite support where distances are also discrete-valued, this divergence is computed directly from empirical histograms; in continuous settings, one would need to explicitly estimate the density of distances, introducing additional modeling choices. An alternative for such settings is to instead compare the samples generated by the model and the reference in identically seeded reverse diffusion trajectories (Kadkhodaie et al., 2024), thereby isolating the impact of the denoising functions.

**Denoiser-level.** We propose to measure how much the trained and reference denoiser outputs differ when evaluated on the same noisy inputs at fixed diffusion times. This time-resolved analysis allows one to localize the onset of bias along the diffusion trajectory and to focus on regimes where denoising is strongly input-dependent. In addition, it enables conditioning on specific starting points of the backward process, thereby concentrating the comparison on regions where biasing effects are most pronounced.

In practice, the reference against which these observables are evaluated depends on the available information. In controlled settings, such as synthetic data models, one may have direct access to the ground-truth distribution or to exact posterior means, enabling direct comparison between model outputs and oracle quantities. In realistic settings with finite data, we use a surrogate reference obtained through a *sample-split analysis*: two models are trained separately on independent subsets of data (Kadkhodaie et al., 2024). If both models remain in an unbiased regime, their predictions are expected to converge toward the same population-level behavior (Favero et al., 2025b). Growing discrepancies between the models signal data-dependent bias, even if test losses continue to improve.

## 3. Experiments on real data

**Experimental setup.** We follow the numerical setup of Bonnaire et al. (2025) on the CelebA face dataset (Liu et al.,

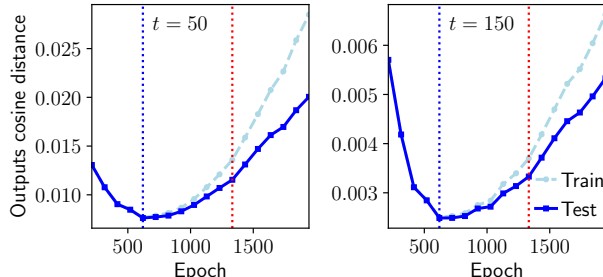

*Figure 2.* Cosine distance between the predictions of two networks trained on disjoint subsets of CelebA of size $n = 1024$, evaluated on inputs noised until time $t$ out of $T = 1000$. The distance is evaluated for original images that are either outside of both training sets ("Test") or in *one of them* ("Train"). The vertical dashed lines correspond to the minima of the diffusion time-averaged metrics shown in Fig. 1(a).

2015), which we convert to grey-scale downsampled images of size $32 \times 32$. The generative process is the DDPM described above, with a U-Net architecture (Ronneberger et al., 2015) trained to predict additive Gaussian noise $\hat{\boldsymbol{\xi}}_t$ from the noised input $\boldsymbol{x}_t$. We provide implementation details in Appendix B.

**Results.** We first measure bias at the sample level. Following identical noise trajectories, we generate samples from models trained on distinct datasets of size $n = 1024$ and quantify their divergence using the cosine distance $D_C = 1 - S_C$, where $S_C(\boldsymbol{x}, \boldsymbol{y}) = \frac{\boldsymbol{x} \cdot \boldsymbol{y}}{\|\boldsymbol{x}\| \|\boldsymbol{y}\|}$. This *sample-split analysis*, averaged over all pairings between 15 trained models, is shown in Fig. 1(a). Despite substantial variability across runs (light shaded curves), the mean exhibits a clear U-shape, with a minimum reached early in training. The DSM test loss, also shown in Fig. 1(a), follows a similar trend but reaches its minimum significantly later. This separation precisely identifies the *biased generalization* phase. Note that the standard indicator of memorization defined in Eq. (3) records a maximum average value across checkpoints below $10^{-3}$.

We illustrate the bias effect qualitatively using representative generated samples in Fig. 1(a) (right; see Appendix B. for details). Up to the maximum similarity point (third row of samples), the two models generate nearly identical images (central columns). Near the minimum of the test loss, the samples develop marked differences and can exhibit features closely resembling their respective nearest training examples (side columns), without exact replication.

We further validate this behavior at the denoiser output level. Fig. 2 shows the cosine distance between the noise predicted by two models evaluated at fixed diffusion times, with the minima of the sample-level bias metric and of the test loss indicated by vertical dashed lines. Importantly, this normalized metric is able to quantify a misalignment between the two models already during the early transient in training,

when both produce outputs of small magnitude and the generative process is almost entirely driven by noise. Focusing on short ($t = 50$) and intermediate ($t = 150$) diffusion times, where denoising is most input-dependent, we observe that denoiser divergence starts increasing close to the onset of sample-level bias, again well before the test loss minimum. Additional evidence is obtained by conditioning on the origin of the noisy input: while the prediction distance is independent of whether inputs originate from the training or test sets in the unbiased phase, a clear separation emerges after the bias minimum. At the test-loss minimum, the gap between training- and test-conditioned curves confirms the presence of training-data bias in the learned denoisers.

## 4. Biased generalization in a controlled setting

### 4.1. Data model

To verify and better understand the biased generalization that we have observed on real images, we now consider a controlled hierarchical data setting with explicit compositional structure, inspired by recent work on context-free grammars (Zhao et al., 2023; Allen-Zhu & Li, 2023; Garnier-Brun et al., 2025; Cagnetta et al., 2024). In particular, we generate discrete sequences, $s^\mu \in \{1, \ldots, q\}^N$ via a tree-based graphical model, characterized by unambiguous production rules $a \rightarrow bc$ with sparse, log-normally distributed weights (more details in Appendix C). The generation process is repeated over $\ell$ layers, leading to sequences of size $N = 2^\ell$. Unless indicated otherwise, we use $\ell = 4$ layers and $q = 6$.

To apply the formalism of continuous diffusion to discrete data, we follow Li et al. (2022) and one-hot encode the sequences $x_0^\mu = \mathrm{onehot}_q(s^\mu)$, where $x_0^\mu \in \mathbb{R}^d$ and $d = Nq$. Conversely, at $t = 0$, the continuous output of the reverse diffusion process is mapped back to a valid clean sample by applying a per-symbol $\arg\max_q$ followed by one-hot encoding in order to generate discrete sequences. At finite $t > 0$, the output of a trained denoiser approximating the true posterior mean $\mathbb{E}(x_0 \mid x_t)$ can be interpreted as a per-site marginal of the posterior $\mathbb{P}(x_0 \mid x_t)$, i.e. a probability distribution over the $q$ possible values of the corresponding discrete symbol. Accordingly, throughout this section the comparison between two posterior-mean estimates is performed by computing the $q$-way KL divergence between predicted and reference one-site marginals at each position $i$, and averaging across the $N$ positions.[1]

**Exact inference and hierarchical filtering.** The tree structure of the data generative model, sketched in Fig. 3(a),

[1]While this averaging of the $D_{\mathrm{KL}}$ implies a factorized representation across sites, we emphasize that it enters only at the level of the evaluation observable, not as an independence assumption in the underlying inference.

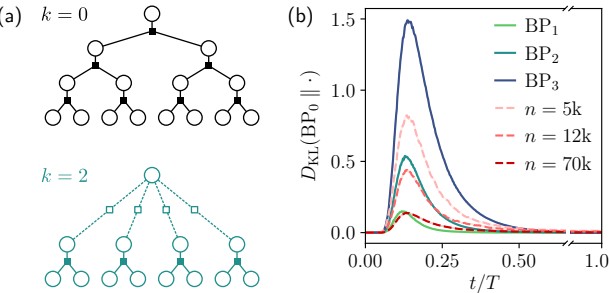

*Figure 3.* (a) Illustration of an $\ell = 3$ tree-based data generation for the full model (top) and a hierarchically filtered version with $k = 2$ (bottom). (b) Kullback-Leibler divergence between the exact posterior mean of $BP_0$ and those from filtered $BP_k$ denoisers along a reverse trajectory following the exact BP, averaged over 2k realizations. Dashed lines show trained models at the minimum of their test losses for different training set size $n$.

has two major advantages compared to real data. Firstly, it enables straightforward i.i.d. sampling through a Markov chain initialized at the root of the tree. Secondly, it allows exact inference of the posterior mean given a noisy observation, $\hat{x}_0(x_t)$, via a dynamic programming algorithm known as Belief Propagation (BP) (Mézard & Montanari, 2009). In addition, the hierarchical structure of the model makes it possible to perform controlled filtering of correlations, following the procedure introduced by Garnier-Brun et al. (2025). By treating the nodes at an intermediate layer of the tree, say at level $k < \ell$, as conditionally independent given the root (Fig. 3), one can selectively suppress long-range correlations in the data. In particular, high-level features involving contiguous blocks of symbols of size $2^{\ell-k}$ or greater are filtered out. The exact BP algorithms associated with these filtered topologies, denoted $BP_k$, can be interpreted as performing increasingly coarse inference over the original hierarchical data. These tools will be used to characterize the emergence of structural scales along the reverse diffusion process and to identify which features are learned by a trained model. Additional details are provided in Appendix C.

**Experimental setup.** Building on the findings of Garnier-Brun et al. (2025), we adopt a vanilla transformer encoder architecture (Vaswani et al., 2017) for the denoiser, which is able to implement a close approximation of BP in this data model. In our context, the model output is transformed via a $q$-wise softmax, allowing $\hat{x}_0^\theta(x_t)$ to be interpreted as a marginal probability distribution over the discrete symbols. We therefore train the model by minimizing the cross-entropy loss between these normalized outputs and the one-hot encoded noise-free original training samples. We use $T = 500$ diffusion steps and a standard linear noise schedule. More details of our implementation and training algorithm can be found in Appendix C.

## 4.2. Dynamical emergence of bias

We focus our denoiser-level comparisons on regions of diffusion time where reverse trajectories produced by different denoisers are most sensitive to the structural properties of the data, and where bias can therefore emerge most clearly. Following Biroli et al. (2024), we can delineate distinct dynamical regimes along the reverse diffusion process. In Fig. 3(b), we report the Kullback–Leibler divergence between the outputs of filtered ($BP_k$) and trained denoisers with the full-tree oracle $BP_0$, evaluated on noisy observations of i.i.d. data. Three regimes can be identified:

*(i) Long-time unstructured regime.* For large diffusion times, $t/T \gtrsim 0.5$, the signal contained in $\boldsymbol{x}_t$ is negligible and all denoisers output the unconditional marginal probabilities $\mathbb{P}(\boldsymbol{x}_0)$. As a result, the filtered and trained denoisers are nearly indistinguishable from the exact oracle, and no sign of bias can be detected. This long-time *unstructured regime* is analogous to phase I identified in Biroli et al. (2024).

*(ii) Intermediate structured regime.* At intermediate times, $0.08 \lesssim t/T \lesssim 0.5$, the outputs of $BP_k$ and of trained denoisers progressively depart from the exact $BP_0$ posterior, reflecting the increasing relevance of longer-range correlations in $\boldsymbol{x}_t$ for the oracle prediction. In line with Garnier-Brun et al. (2025), increasing the training set size allows trained denoisers to resolve progressively finer structure (pink–red dashed lines) and decrease the distance to the oracle. For all imperfect denoisers, the KL distance to the exact posterior mean peaks around a critical time $t/T \approx 0.15$, consistent with the observations of Sclocchi et al. (2025a) in discrete diffusion models. We therefore select this critical point for fixed-time denoiser comparisons (Fig. 1(b), Fig. 6).

*(iii) Short-time trivial regime.* At short diffusion times, $t/T \lesssim 0.08$, all denoisers again appear effectively equivalent. In this regime, the noise level is sufficiently low that denoising i.i.d. sequences can be achieved by simple symbol-wise rounding, without exploiting any correlation structure, a behavior specific to the discrete data setting. Note that differences between denoisers would instead emerge when considering neighborhoods of sequences that are out-of-distribution for the oracle denoiser but remain admissible for models that ignore long-range correlations. However, because the supervision signal during training is restricted to weakly noised versions of valid training samples, it provides no information that would distinguish the exact (or empirical) posterior mean from such a naive symbol-wise prior in this regime. As a result, even imperfectly trained denoisers converge to the simplest explanation compatible with the loss and exhibit identical behavior at short times (see Fig. 12 in Appendix D), rendering this regime uninformative for probing bias.

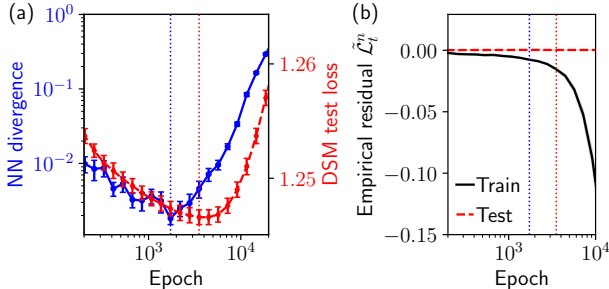

*Figure 4.* Bias metrics for a diffusion model trained on $n = 12$k tree-based sequences computed with 50k generated samples and evaluation points, and averaged over 15 training runs and showing the standard error. (a) Nearest-neighbor divergence (Sec. 2.2) of generated sequences and denoising test loss as a function of training. (b) Expectation value of the excess data-dependent loss of (4) evaluated on test and train sequences noised up to time $t = 150$. Vertical dashed lines show the minimum of the bias metric (blue) and test loss (red).

## 4.3. Early-stopped models are biased

**Sample-split analysis.** As in Sec. 3, we independently train two denoisers on disjoint subsets of the data and monitor the model divergence, quantified by the $D_{\mathrm{KL}}$ between their predicted posterior means on identical noised inputs. The key question is when this divergence emerges relative to standard generalization metrics. In the present controlled setting, access to the exact posterior mean via BP allows us to additionally track the $D_{\mathrm{KL}}$ between each trained model and the oracle posterior mean. The minimum of this quantity marks the point of closest approximation to the true posterior and provides a sample-efficient estimate of the test-loss minimum (see loss decomposition below). As shown in Fig. 1(c), the onset of model divergence occurs *before* this minimum: the models enter a data-dependent regime while still improving toward the exact posterior mean, thereby confirming the existence of a biased generalization phase.

**Nearest-neighbor divergence.** We next examine the distribution of nearest-neighbor overlaps—here the fraction of symbols in agreement between two sequences—between generated samples and the training data, taking advantage of our ability to sample i.i.d. from our data model. Fig. 4(a) shows the nearest-neighbor divergence as a function of training, displaying a minimum reached significantly before the minimum of the DSM test loss. The training epoch at which this metric begins to increase closely matches the onset of model divergence identified by the sample-split analysis in Fig. 1(c), confirming the consistency of the two diagnostics of generative bias. The reported curves are averaged over 15 independently trained models, as the adaptive optimizer can induce abrupt fluctuations in this extreme-value statistic along training; see Appendix D.2.

**Identifying bias with a loss decomposition.** With access to the exact posterior mean, we derive and evaluate an explicit decomposition of the denoising loss that makes the

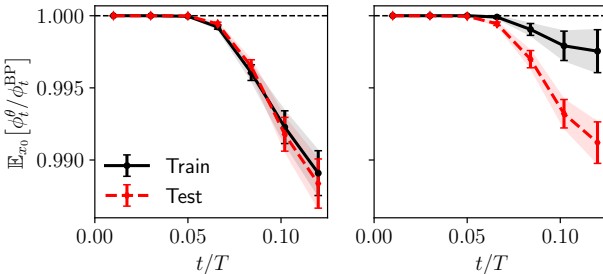

*Figure 5.* Average normalized overlap to the starting sequence for a "U-turn" experiment after noising to time $t$ using 100 trajectories for each 1000 starting points, model trained on $n = 12$k data. Left: checkpoint minimizing the Kullback-Leibler divergence of nearest neighbor overlaps with the result expected from fair sampling. Right: checkpoint minimizing the denoising test loss. Shaded areas show the standard error over starting sequences.

buildup of data bias transparent. Each cross-entropy loss term can be rewritten as

$$
\ell_t(\theta, \boldsymbol{x}_0, \boldsymbol{x}_t) = \overbrace{-\hat{\boldsymbol{x}}_0(\boldsymbol{x}_t)^\top \log \hat{\boldsymbol{x}}_0^\theta(\boldsymbol{x}_t)}^{\text{Exact distillation } \ell_t^\star} \\
\underbrace{- (\boldsymbol{x}_0 - \hat{\boldsymbol{x}}_0(\boldsymbol{x}_t))^\top \log \hat{\boldsymbol{x}}_0^\theta(\boldsymbol{x}_t)}_{\text{Excess data-dependent } \tilde{\ell}_t}, \quad (5)
$$

where the logarithm is taken element-wise. The test denoising loss is obtained by taking the expectation $\mathbb{E}_{\boldsymbol{x}_0, \boldsymbol{x}_t}[\ell_t(\theta, \boldsymbol{x}_0, \boldsymbol{x}_t)]$ over the joint distribution of $(\boldsymbol{x}_0, \boldsymbol{x}_t)$ induced by the draw of data and diffusion noise. For fixed model parameters $\theta$, we have $\mathbb{E}_{\boldsymbol{x}_0}\left[ \tilde{\ell}_t(\theta, \boldsymbol{x}_0, \boldsymbol{x}_t) \mid \boldsymbol{x}_t \right] = 0$, and therefore, by the law of total expectation, $\mathbb{E}_{\boldsymbol{x}_0, \boldsymbol{x}_t}[\tilde{\ell}_t] = 0$. As a consequence, the excess term, measuring data-dependent overconfidence, does not contribute on average to the test loss, as verified in Fig. 4. In contrast, for the training loss the expectation over $\boldsymbol{x}_0$ is replaced by the empirical average over the training set. In this case, the excess loss, quantifying the mis-calibration of the model relative to the exact posterior mean induced by the pull of the empirical distribution, is finite and can be optimized at training. As shown in Fig. 4(b), this term starts decreasing significantly in the biased-generalization phase. Crucially, before the minimum of the test loss, the model is able to simultaneously optimize *both* the distillation term and the excess data-dependent term, providing a concrete illustration of orthogonality and coexistence of generalization and memorization.

**Bias in the reverse process.** We further probe the *dynamical* emergence of bias along diffusion trajectories by means of a "U-turn" experiment. Starting from a clean sample $\boldsymbol{x}_0 \sim P_0$, we noise it up to diffusion time $t$. We then follow the reverse *process* using a given denoising posterior mean, yielding a reconstructed sample $\tilde{\boldsymbol{x}}_0$ at $t = 0$. Bias can be detected by comparing the outcomes of such U-turns when conditioned on starting points drawn from the training set

versus previously unseen data, using BP as a reference for unbiased behavior.

For each fixed starting point $\boldsymbol{x}_0$, we compute the overlap between $\tilde{\boldsymbol{x}}_0$ and $\boldsymbol{x}_0$ over multiple U-turn realizations, using either the trained model or BP as the denoising posterior mean. This defines the average overlap $\phi_t^\theta(\boldsymbol{x}_0)$ and $\phi_t^{\text{BP}}(\boldsymbol{x}_0)$, see Appendix C for a more detailed definition. We then consider the ratio $\phi_t^\theta(\boldsymbol{x}_0)/\phi_t^{\text{BP}}(\boldsymbol{x}_0)$, which isolates deviations of the trained model from optimal (oracle) behavior, and finally average this quantity over starting sequences.

Figure 5 shows the evolution of $\mathbb{E}_{\boldsymbol{x}_0}[\phi_t^\theta/\phi_t^{\text{BP}}]$ for a trained model ($n = 12$k) at the checkpoint minimizing the nearest-neighbor divergence and at a later checkpoint minimizing the test loss. At the former, the curves corresponding to training and test starting points are indistinguishable within statistical fluctuations. At the test-loss minimum, by contrast, a clear separation emerges: corrupted training samples are significantly more likely to be recovered by the reverse process than previously unseen data, providing direct dynamical evidence of training-data bias. At the same time, both curves move closer to unity, corresponding to oracle performance, even on the test set. This illustrates once again how the test loss can continue to decrease despite the emergence of bias.[2] We show a more complete evolution of such U-turn experiments along training and for another trained model in Appendix D.2.

## 5. Understanding the emergence of bias

### 5.1. Sequential learning and bias

Both in real data and in the hierarchical data setting, we have shown that *biased generalization* does not emerge immediately during training, nor exactly at the test loss minimum. We now discuss the underlying mechanism, which can be understood in detail in our controlled setting, refining the description provided in (Favero et al., 2025b).

**Bias mechanism.** In the context of tree-based data models, the $\ell$ hierarchical levels of features to be learned induce a sequential, staircase-like discovery process (Garnier-Brun et al., 2025; Favero et al., 2025a). On the one hand, for a given number of training samples $n$, the network has sufficient statistics to reliably learn only the first $\ell - k^\star(n)$ levels of the hierarchy, with $0 \leq k^\star(n) \leq \ell$. On the other hand, along the training dynamics, the model learns features in order of increasing complexity. While the model is focusing on simpler, shorter-range features corresponding to layers $k > k^\star(n)$, it follows a gradient signal that is effectively indistinguishable from that of the population loss in the early dynamics, exhibiting a "mean-field" feature learning regime (Montanari & Urbani, 2026). As long as the network

---

[2]The ratio remains below one because the trained model only partially approximates the BP denoiser; see Sec. 5.1.

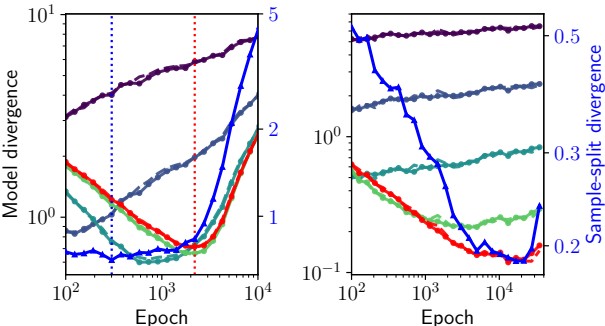

*Figure 6.* Evolution of model divergences $D_{\mathrm{KL}}(\cdot \;\|\; \hat{\boldsymbol{x}}_0^\theta)$ for different $\mathrm{BP}_k$ (from purple to green, decreasing in $k$) and $\mathrm{BP}_0$ (red), evaluated on models A ($-\bullet$) and B ($--$). Blue curves ($-\blacktriangle$) show the sample-split denoiser divergence $D_{\mathrm{KL}}(\hat{\boldsymbol{x}}_0^{\theta_A} \;\|\; \hat{\boldsymbol{x}}_0^{\theta_B})$. Left: $n = 5\text{k}$, Right: $n = 70\text{k}$.

remains in this *partial generalization* regime (Favero et al., 2025b), its representation is approximately unbiased, and the outputs of models trained on different data splits remain aligned. If one trains the model further, the training data are insufficient to fully resolve the next level, so the models adopt a data-biased approximation of it, resulting in a simultaneous decrease in the test loss and an increase in any of the considered bias measurements. We have shown that the onset of this biased phase occurs before the minimum of the test loss.

**Empirical validation.** We verify this picture by leveraging the $\mathrm{BP}_k$ descriptions (Sec. 4.1), i.e., oracle denoisers that resolve exactly $\ell - k$ levels of the hierarchy. In Fig. 1(b), we show the $D_{\mathrm{KL}}$ between each $\mathrm{BP}_k$ and the transformer denoiser along training, for an intermediate training set size $n = 12\text{k}$. We contrast these curves with the sample-split denoiser divergence, detecting the emergence of bias. As training progresses, the model sequentially approaches $\mathrm{BP}_k$ with decreasing $k$, reflecting resolution of longer-range correlations. For this value of $n$, the best match is achieved with $\mathrm{BP}_1$, indicating that the model cannot fully resolve the highest level of the hierarchy and thus $k^\star = 1$. Beyond the maximal similarity to $\mathrm{BP}_1$, the model continues to improve towards the ground truth while bias starts increasing. In Fig. 6, we show analogous plots for smaller ($n = 5\text{k}$) and larger ($n = 70\text{k}$) training sets. With fewer samples, the best match occurs at a higher $k^\star = 2$, and the biased region is wider; with more samples, the model approaches $\mathrm{BP}_0$, and the bias becomes negligible. Importantly, in all cases the onset of bias follows the epoch at which the best-matching $\mathrm{BP}_{k^\star}$ is reached. We note that the sequential learning dynamics we describe is not perfectly rigid: the transitions between levels are gradual rather than sharp, particularly when using adaptive optimizers such as Adam. Nonetheless, the overall picture of progressive feature learning followed by data-dependent approximation of finer structure provides a coherent explanation for the emergence of biased generalization.

Such a clear hierarchy of features is of course inaccessible directly in real data. As a complementary image-domain diagnostic, we therefore repeat the comparison on the CelebA images of Sec. 3 after constructing filtered datasets either from leading principal components or from a Haar wavelet decomposition, see Appendix B.3. While the PCA basis keeps the corresponding U-Net noise-prediction similarities strongly compressed across levels, and therefore does not appear to extract features relevant to the learning dynamics, the wavelet basis exposes a clearer ordering of coarse and finer structural image scales. This behavior appears consistent with the mechanism identified in the controlled model. A natural way to sharpen this analysis further would be to extract a controllable amount of features using nonlinear networks, in the spirit of Refinetti et al. (2023) and Gerace et al. (2023), which we leave for future work.

### 5.2. A training-free example of biased generalization

The sequential discovery of features plays a key role in locating the biased generalization phase in our trained model, yet this phenomenon is not unique to neural network denoisers and can arise in much simpler settings. We now consider a single-parameter probability distribution focused on the training data, still sampled i.i.d. from the tree-generative model. The parameter $\varepsilon$, which we refer to as the *sharpness* of the distribution, controls the concentration of the probability mass on the data, by assigning probability weight $\propto 1$ to the correct symbols, and a uniform weight $\propto e^{-\varepsilon}$ to all the others. The associated denoising posterior mean can thus be interpreted as a smoothed version of the empirical posterior mean—exactly recovered in the limit $\varepsilon \to \infty$, while the $\varepsilon \to 0$ limit yields a uniform distribution. Full details are reported in Appendix C.5.

We can use this parametric estimator $\tilde{P}_0^\varepsilon$ in generative diffusion by exactly computing the associated posterior means, and employing them in the reverse process. Thus, we can monitor the DSM test loss as a function of $\varepsilon$, while efficiently generating samples to detect bias. In Fig. 1(a), we plot the nearest-neighbor divergence alongside the test loss, as a function of the sharpness $\varepsilon$. Consistent with our findings on trained models, we detect the existence of a wide phase of *biased generalization*, suggesting that a misalignment between distribution matching and sample fairness objectives might emerge beyond the specific settings analyzed in this paper. Comparing these results with the trained transformer of Sec. 4.3, the bias is markedly more severe in this training-free model. Indeed, in the absence of any sequential feature learning, the smoothed posterior mean concentrates probability mass around the training samples directly, with no intermediate phase in which a hierarchy of universal features can be extracted before data-dependent bias sets in. The trained networks therefore enjoy a partial protection from bias that the architecture-free model

lacks, even as the underlying tension between distribution matching and sample fairness remains. Note that a high-dimensional counterpart to this training-free example was analyzed in (Biroli & Mézard, 2026), where the optimal width (that minimizing the KL divergence with $P_0$) in a simple Kernel Density Estimation problem was located beyond a phase-transition where the shape of the estimator becomes strongly data-dependent. This exactly solvable example of biased generalization thereby demonstrates that the phenomenon may arise even in asymptotic regimes.

## 6. Limitations

Our study focuses on the standard DDPM setting, and does not address whether analogous biased regimes arise with other classes of modern generative models. In the context of the discrete hierarchical model that we use, it would be particularly relevant to assess whether more specialized generative processes, such as discrete diffusion or other autoregressive models, could perhaps mitigate the effect. Similarly, while we provide evidence on real images and in a controlled setting, our empirical evaluation is also necessarily limited to moderate-scale datasets and architectures, and we do not explore the behavior of large-scale diffusion models trained on higher dimensional data. A related open question is how the boundaries of the biased-generalization regime scale with the data dimension $d$ and the training set size $n$. More precisely, for a given data distribution and model, what is the range of training, if any, in which some structural scales are resolved while others are not? For any structured distribution with multi-scale features, we expect such a regime to exist for appropriate values of $d$ and $n$, but its precise extent should be data- and model-dependent, and understanding how to reduce it appears to be a relevant direction to explore.

Moreover, in this work, the purpose of our analysis is primarily diagnostic: we do not propose mitigation strategies or alternative training objectives designed to prevent it. Finally, bias is quantified through proximity-based and denoiser-based metrics, which cannot capture semantic similarity or downstream perceptual notions of memorization.

## 7. Conclusion

In this work, we identify and characterize a regime of *biased generalization* in generative diffusion models, in which the test loss continues to decrease while the generative behavior of the model becomes increasingly dependent on the specific training samples. At a conceptual level, this phenomenon reflects a simple but important fact: minimizing a distributional discrepancy between a model and the data-generating process does not, by itself, preclude biased sampling toward the finite dataset used for training. We document this effect

across qualitatively different settings, spanning continuous images with convolutional U-Nets, discrete hierarchical sequences with transformer architectures, and a training-free architecture-free estimator, indicating that biased generalization is not tied to a specific data modality, architecture, or optimization procedure. While related observations had previously been made in other generative settings (van den Burg & Williams, 2021; Biroli & Mézard, 2026), we show that this effect arises naturally and universally in diffusion models. By leveraging a controlled hierarchical data model, we trace the early emergence of bias to the sequential nature of feature learning in deep networks (Refinetti et al., 2023; Bardone et al., 2025), whereby increasingly fine structure is approximated in a data-dependent manner once available statistics become insufficient. As a result, biased generalization can arise *before* the test loss minimum, highlighting a limitation of early stopping as a safeguard against training-data bias. These observations have practical implications for the evaluation and deployment of generative models. Implicitly conflating generalization with unbiased sampling could become particularly consequential as generative models are deployed in domains where privacy, memorization, and data provenance are critical. Beyond the issue of training-data bias, defining appropriate metrics to quantify (or even clearly define) generalization remains a challenge in self-supervised learning, see e.g. Mendes et al. (2026).

An important open direction is to understand how common generation mechanisms may interact with the biased generalization regime. In particular, conditioning and guidance techniques such as classifier-free guidance (Ho & Salimans, 2022) may amplify subtle data-dependent biases by selectively steering generation toward features overfitted to specific training data (Carlini et al., 2023; Somepalli et al., 2023). Clarifying the interplay between biased generalization, conditioning, and data extraction is a promising direction for future work.

## Acknowledgements

The authors are grateful to G. Biroli, S. Moran, P.-F. Urbani and R. Urfin for stimulating discussions and to E. Moscato who participated in the early stages of this project. The work of J. G.-B. was supported by the European Union's Horizon Europe program under the Marie Skłodowska-Curie grant agreement No. 101210798.

## Reproducibility statement

We provide the source code used to perform our numerical experiments on the repository accessible at https://github.com/davide-beltrame/biased-generalization. It includes a Python script generating the hierarchical data, as well as the PyTorch

implementation of our numerical experiments, as well as an efficient implementation of the Bayes-optimal denoising. The hierarchical data used to produce the figures in the main text corresponds to fixing `seed = 0`, `q_eff = 4` and `sigma = 1` in the data generation script, see Appendix C for details on the role of the latter two parameters.

## Impact Statement

This work aims to improve the understanding of generalization in modern generative diffusion models by identifying and characterizing a regime in which models exhibit systematic bias toward their training data despite continued improvement in standard test metrics. By providing quantitative tools to detect such biased behavior, our results have potential positive implications for applications where data privacy, copyright, or unintended data reuse are concerns.

More broadly, our findings highlight a limitation of common evaluation practices based solely on held-out performance or sample quality, and suggest that additional care may be required when deploying generative models trained on sensitive or proprietary datasets. We view this contribution as complementary to ongoing efforts to develop more robust, transparent, and privacy-aware generative modeling methodologies. We do not anticipate immediate negative societal consequences arising from this work.

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

# A. Extended related works

**Memorization vs. overfitting in generative models.**    van den Burg & Williams (2021) show that VAEs can remain sensitive to their training set without verbatim replication, a conceptual precursor to our findings in a different model class and without controlled analysis of the training dynamics. Yoon et al. (2023) argue that memorization and generalization are mutually exclusive in diffusion models, a view that our results directly challenge. Chae et al. (2025) study benign overfitting in overparameterized generative models; our results suggest caution with this framing, as any train/test discrepancy entails data-dependent behavior.

**Generalization-to-memorization transition.**    A growing body of work locates the onset of memorization in diffusion models at or after the test-loss minimum (Li et al., 2023; George et al., 2026; Favero et al., 2025b; Bonnaire et al., 2025; Montanari & Urbani, 2026). Our contribution is complementary, in that we show that data-dependent bias appears strictly earlier, while the test loss is still decreasing, and provide a quantitative framework to detect it.

**Memorization detection and privacy.**    Several recent works develop tools to detect or characterize memorization in trained diffusion models (Gu et al., 2025; Carlini et al., 2023; Somepalli et al., 2023; Ross et al., 2025). Our sample-split diagnostics complement these by exposing a softer form of bias that precedes overt memorization and is invisible to nearest-neighbor metrics evaluated on a single model.

**Sequential learning dynamics.**    A parallel line of work characterizes the training dynamics of diffusion and related generative models in tractable regimes. In the linear/Gaussian regime, the dynamics decouple in the eigenbasis of the data covariance, and learning proceeds from leading to subleading eigendirections (Catania et al., 2025; Merger & Goldt, 2025; Wang & Pehlevan, 2026); the placement of optimal early stopping can then be related to the spectral hierarchy. Beyond second-order statistics, deep networks have been shown to acquire features sequentially across timescales, learning coarse, low-order structure before finer, higher-order one (Refinetti et al., 2023; Bardone et al., 2025; Garnier-Brun et al., 2025). Our mechanism in Sec. 5.1 builds on this stronger notion, since bias toward individual training samples arises specifically when finer hierarchical features begin to be resolved in a data-dependent manner, a regime beyond what spectral analyses can capture.

**Test loss as a generalization metric.**    Concurrently and independently to our submission, Mendes et al. (2026) show in a solvable autoencoder model that decreasing test loss can coexist with degrading representation quality, conceptually echoing our findings. More recently, Marion & Wu (2026) argue that the field should pivot to studying the pre-memorization phase and reports unexplained double-descent phenomena in distributional distances that are invisible to train-test gaps; the diagnostics we develop here apply directly to that regime. More fundamentally, Biroli & Mézard (2026) prove in an exactly solvable high-dimensional KDE problem that even optimizing the KL divergence does not preclude bias toward training samples beyond a phase transition, supporting the broader relevance of our conclusions.

# B. Additional details on CelebA experiments

### B.1. Experimental setup

Our experiments leverage the codebase of Bonnaire et al. (2025), available at `https://github.com/tbonnair/Why-Diffusion-Models-Don-t-Memorize`, and follow closely their setup, see `https://github.com/davide-beltrame/biased-generalization` for our slightly modified version.

**Dataset.**    We use the CelebA dataset (Liu et al., 2015), center-cropped, gray-scaled and resized to $32 \times 32$ pixels. No data augmentation is used. All the considered models are trained of different subsets of size $n = 1024$ of the original dataset.

**Neural network architecture and diffusion.**    We use a standard U-Net architecture (Ronneberger et al., 2015) with base channel width 32 and channel multipliers $(1, 2, 4)$. Self-attention is applied at the $16 \times 16$ and $8 \times 8$ resolution levels (i.e. the second and third encoder/decoder blocks). The network is used within the standard DDPM (Ho et al., 2020) framework and is trained to predict additive Gaussian noise across $T = 1000$ steps with a linear noise schedule for $\beta_t$, ranging from $10^{-4}$ to 0.02.

**Training and generation.** All models are trained with the Adam (Kingma & Ba, 2014) optimizer, with a learning rate $\eta = 1 \times 10^{-4}$, $(\beta_1, \beta_2) = (0.9, 0.999)$ and a batch size of 512. Training time is measured in epochs, where within each epoch each training point is corrupted according to Eq. (1), with a timestep $t$ drawn uniformly at random in $[1, 1000]$. All our models are trained for 5000 epochs.

To generate images from a trained model, we follow the standard DDPM procedure: starting from white noise, we repeatedly compute:

$$\boldsymbol{x}_{t-1} = \frac{1}{\sqrt{\alpha_t}} \left( \boldsymbol{x}_t - \frac{1 - \alpha_t}{\sqrt{1 - \bar{\alpha}_t}} \hat{\boldsymbol{\xi}}_\theta(\boldsymbol{x}_t) \right) + \sqrt{\beta_t} \boldsymbol{z}, \qquad \boldsymbol{z} \sim \mathcal{N}(0, \mathbf{I}). \tag{6}$$

for $t = T, ..., 1$, where $\hat{\boldsymbol{\xi}}_\theta(\boldsymbol{x}_t)$ is the output of the model given $\boldsymbol{x}_t$.

### B.2. Denoiser-level experiments

As mentioned in the main, for CelebA we do not have access to the ground truth. For this reason, we perform a sample-split analysis by comparing the predicted noises of two models (A and B) trained on different subsets of the original dataset. We train both models for the same number of epochs and compare them at corresponding training checkpoints. Overall, we train 15 models, resulting in 105 different model pairs. Our results are averaged over all such pairs.

For the denoiser-level analysis presented in Fig. 2, we adopt the following procedure. First, we select a certain time-step $t \in [1, 1000]$. Then, we select $N_{\text{eval}} = 1000$ training and test images and we apply Eq. (1) to each of them, resulting in $N_{\text{eval}}$ noisy images per split. Notice that the training points belong to dataset A, hence effectively representing test points for model B. The test points are taken from a third dataset different from both A and B. Finally, we compute the cosine similarity between the noise predictions of the two models when given the same noisy image as input:

$$\text{sim}(\boldsymbol{x}_t) = C_S(\hat{\boldsymbol{\xi}}_{\theta_A}(\boldsymbol{x}_t), \hat{\boldsymbol{\xi}}_{\theta_B}(\boldsymbol{x}_t)). \tag{7}$$

We compute the above quantity for each of the $N_{\text{eval}}$ noisy images and we take the average. We repeat this procedure for all available training checkpoints. The final result shown in Fig. 2 is the mean (with corresponding standard error) computed over all model pairs. We present the results obtained for times $t = 100, 200$ in Fig. 9, showing the same phenomenology as in Fig. 2.

### B.3. Coarse-to-fine filtering experiment

The controlled experiments in Sec. 4 make the hierarchy explicit by construction, as the filtered $\text{BP}_k$ denoisers are oracle references for different levels of the generative process. CelebA has no such oracle decomposition, therefore we use image-domain filters as a diagnostic probe. Given a proposed decomposition of images into progressively richer levels, we ask whether U-Nets trained on those levels align with an unfiltered U-Net in an ordered way.

We compare two decompositions. In the PCA construction, images are projected onto the leading empirical principal components estimated from the two training splits and then reconstructed. Levels $L0, L1, L2, L3$ retain $32, 128, 512, 1024$ components, respectively. In the Haar wavelet construction, we apply a three-level two-dimensional Haar transform and reconstruct images after removing progressively fewer detail bands: $L0$ keeps only the coarsest approximation, $L1$ and $L2$ add increasingly finer detail bands, and $L3$ is the original image.

For each decomposition and level, we train the same U-Net architecture as in Sec. 3 on two disjoint filtered training splits of size $n = 1024$. Before performing the cross-level comparison, we first choose the checkpoint of each filtered $L0, L1, L2$ reference model using the following "within-level sample-split" criterion. For a fixed level $Lk$, we compare the predicted noises of the two independently trained $Lk$ models at matching checkpoints on 512 held-out $Lk$ inputs noised to diffusion time $t = 100$, and select the checkpoint at which their average cosine similarity is highest. Each simpler filtered model is thus stopped at the point where its two split-trained versions agree most, before it may have developed split-specific bias. These selected checkpoints are shown as diamonds in Fig. 7.

We then freeze the selected $L0, L1, L2$ models and sweep the checkpoints of an unfiltered $L3$ (full data) model. At each checkpoint, we evaluate all models on the same 512 held-out $L3$ images noised to diffusion time $t = 100$ and compute the cosine similarity between their predicted noises. This amounts to asking which filtered reference is closest to the unfiltered model at each stage of training.

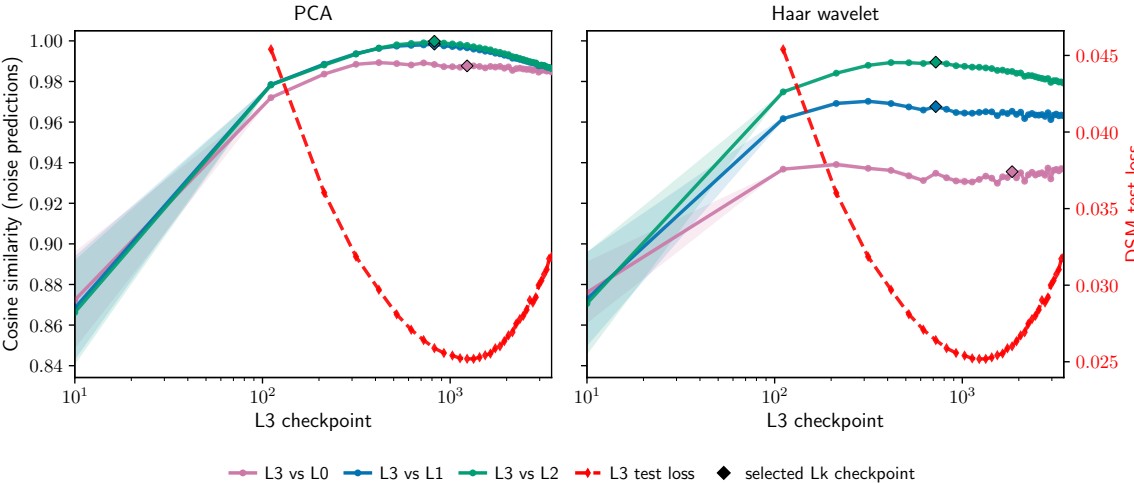

*Figure 7.* Coarse-to-fine filtering experiment on CelebA. Left: PCA filtering provides a global covariance-based control. The ordering here is weaker as the curves remain strongly compressed, indicating that this basis does not isolate the relevant feature hierarchy as cleanly. Right: Haar wavelet filtering produces a clear multiscale ordering after the early transient. The coarsest reference $L0$ reaches its largest similarity early and then remains approximately stable, while the finer references $L1$ and $L2$ peak later. Similarities are cosine similarities between predicted noises on matched $L3$ test inputs at diffusion time $t = 100$. Diamonds mark the filtered $Lk$ checkpoints selected by the within-level sample-split criterion. Dashed curves show the $L3$ test loss.

In Fig. 7 the PCA panel shows that although there is a weak ordering, the cross-level similarities are all close to one after the early transient, so truncating global principal components does not produce well-separated reference denoisers for this convolutional U-Net. We understand this observation to signal that the principal components are not the relevant features being learned by the model, which is to be expected given the simplicity of such a decomposition. The right panel shows that the Haar wavelet experiment exposes a clearer ordering after the early transient. $L3$–$L2$ similarities remain above $L3$–$L1$, which in turn remain above $L3$–$L0$. The coarsest $L0$ comparison reaches its maximum early and then saturates rather than degrading, suggesting that these coarse features do not show the same data bias as finer references, consistent with our interpretation. Finer comparisons peak later and their decrease is compatible with the onset of bias observed in the sample-split diagnostics of Sec. 3.

### B.4. Sample-level experiments

For the sample-level analysis, we adopt the same sample-split approach as in the denoiser-level analysis. Given a pair of models A and B, for each of them we generate $N_{\text{eval}} = 512$ images starting from random noise. Importantly, following (Kadkhodaie et al., 2024; Favero et al., 2025b), we fix both the starting random noise and the reverse process noise realizations $z$ in Eq. (6). We then compute the cosine similarity between the standard-normalized final images produced by the models as:

$$\text{sim}(\hat{\boldsymbol{x}}_0^A, \hat{\boldsymbol{x}}_0^B) = C_S(\hat{\boldsymbol{x}}_0^A, \hat{\boldsymbol{x}}_0^B). \tag{8}$$

We average this quantity over $N_{\text{eval}}$ generated images per model and we repeat this procedure across checkpoints. Finally we compute the mean and standard error across all 105 different model pairs, resulting in Fig. 1(a).

For each model pair and for each checkpoint, we hence obtain an empirical distribution of cosine similarities built from the $N_{\text{eval}}$ pairs of generated images. In order to select the images reported in the right panel of Fig. 1 (a), we consider the top 20 pairs whose cosine similarity is closer in absolute value to the mean of the aforementioned distribution (see Fig. 8). The images in Fig. 1(a) are extracted from such 20 pairs at different checkpoints. The closest training images to the side of each generated sample are obtained through the same approach as in Eq. (8).

## C. Additional details on controlled experiments on hierarchical data

We provide the source code used to perform our numerical experiments on the repository accessible at `https://github.com/davide-beltrame/biased-generalization`. It includes a Python script generating the hierarchical data,

as well as the PyTorch implementation of our numerical experiments, as well as an efficient implementation of the various BP denoising.

## C.1. Data model and training

**Dataset generation.** Following Garnier-Brun et al. (2025), we generate discrete sequences, $s^\mu \in \mathcal{X} = \{1, \ldots, q\}^N$ through a tree-based graphical model, specified by a transition tensor $\mathsf{M} \in \mathbb{R}_+^{q \times q \times q}$, known as the "grammar". The grammar assigns probabilities $M_{abc}$ to all possible production rules $a \to bc$. The generation process is then repeated over $\ell$ layers, leading to sequences of size $N = 2^\ell$. This framework is closely linked to the random hierarchy model (Cagnetta et al., 2024), and in general to context-free grammars traditionally studied in the context of natural language processing (Zhao et al., 2023; Allen-Zhu & Li, 2023). In our study, we focus on grammars with distinct production rules, $\mathbf{M}_a \in \mathbb{R}_+^{q \times q}$, for each ancestor symbol $a$, such that $M_{abc} M_{a'bc} = 0 \, \forall a' \neq a$. This choice ensures that the data-generating process is unambiguous, i.e., that ancestor reconstruction from an entire sequence is deterministic. Moreover, we induce sparsity by fixing the number of non-zero elements in $\mathbf{M}_a$ to $\tilde{q} < q$. The non-zero probabilities are drawn from a log-normal distribution of log-mean 0 and log-scale $\sigma$ before being normalized.

In the results presented above, we use a vocabulary size $q = 6$, effective $\tilde{q} = 4$ and tree depth $\ell = 4$, yielding sequence length $N = 16$. We report averages over multiple independent training runs and data (e.g. 15 runs for the $n = 12\mathrm{k}$ curves in Fig. 4), always generated with the same fixed grammar $\mathsf{M}$. The exact sequences we used in the experiments shown in this paper may be reproduced by setting `seed=0` in our data generation script.

**Hierarchical filtering.** To selectively suppress long-range correlations, Garnier-Brun et al. (2025) introduces a *filtering parameter* $k \in 0, \ldots, \ell$, which truncates the hierarchy above level $k$. Nodes at depth $k$ are sampled independently conditioned on the root, while the standard branching process applies below that level. Concretely, for each node $x_j$ at depth $k$,

$$\mathbb{P}(x_j = b \mid x_0 = a) = \left( M_{\sigma_0(j)} M_{\sigma_1(j)} \cdots M_{\sigma_{k-1}(j)} \right)_{ab},$$ (9)

where $\sigma_m(j) \in L, R$ denotes the branch taken at level $m$, and the effective transition matrices

$$(M_L)_{ab} = \sum_c M_{abc}, \qquad (M_R)_{ac} = \sum_b M_{abc}$$ (10)

are obtained by marginalizing over one child. This construction preserves all marginals up to scale $2^{\ell-k}$, while eliminating correlations at larger scales. The cases $k = 0$ and $k = \ell$ correspond to a fully hierarchical model and a conditionally independent (naive Bayes) model, respectively. In the diffusion setting of the main text, filtered models define *coarse-grained oracles* that resolve only a subset of hierarchical features, and play a central role in diagnosing which structural scales are captured by a trained denoiser.

**Continuous diffusion for discrete sequences.** We follow the setup proposed in Li et al. (2022): each token is one-hot encoded, $\boldsymbol{x}_{0,i} = \boldsymbol{e}_{s_i} \in \{0, 1\}^q$. We use Gaussian forward diffusion (Eq. (1)) with $T = 500$ steps and we use the schedule $\overline{\alpha}_t = \prod_{i=1}^t (1 - \beta_i)$ with a linear $\beta$ schedule from $\beta_1 = 2 \times 10^{-4}$ to $\beta_T = 4 \times 10^{-2}$.

**Transformer architecture and training.** We use a transformer denoiser trained with the discrete DSM objective (cross-entropy between the true token $s_i$ and the predicted posterior mean at that position) with random timestep $t \sim \mathrm{Unif}(\{1, \ldots, T\})$. The transformer has 8 layers, hidden dimension 512, 4 attention heads, and uses standard positional plus sinusoidal time embeddings. Training is performed with the Adam (Kingma & Ba, 2014) optimizer with learning rate $3 \times 10^{-4}$, batch size 512, cross-entropy loss. Models trained on 5000, 12000 and 70000 training points are optimized for 20000, 30000 and 35000 epochs respectively.

## C.2. Controlled oracles: Belief Propagation

Belief Propagation is a dynamic programming algorithm that relies on message-passing along the edges of the tree to compute posterior distributions for the symbols in the graph, given knowledge of the transition tensor $\mathsf{M}$, and of a prior on the values of the symbols. Note that it is here equivalent to a simplified inside–outside algorithm for a fixed topology tree. BP proceeds by passing messages along tree edges. For a node $u$, messages are probability vectors over $\mathcal{X}$. In the unfiltered

part of the tree, factor-to-variable updates read

$$\hat{\nu}_{u\to p}(x_p) \propto \sum_{x_u, x_{u'}} M_{x_p x_u x_{u'}}, \nu_{u\to\alpha}(x_u), \nu_{u'\to\alpha}(x_{u'}), \tag{11}$$

while variable-to-factor messages are products of incoming messages. Above the filter level, messages involve only unary conditionals $\mathbb{P}(x_j \mid x_0)$ (Garnier-Brun et al., 2025). Given the tree structure of the factor graph, convergence of BP is achieved in a single upward-downward pass. Once converged, the marginal of any node $i$ is given by

$$\mu_i(x_i) \propto \prod_{\alpha \in \partial i} \hat{\nu}_{\alpha\to i}(x_i). \tag{12}$$

In the diffusion experiments of the main text, BP is used to compute the exact conditional posterior mean $\mathbb{E}[x_0 \mid x_t]$ under the full model and its filtered counterparts $\mathrm{BP}_k$. The prior, conditioning on the observation $\boldsymbol{x}_t$ is introduced in the form of a "field" acting on the sequence elements in the direction of a noisy observation,

$$\boldsymbol{h}_t = \operatorname*{softmax}_{q} \left( \frac{\sqrt{\bar{\alpha}_t}}{1 - \bar{\alpha}_t} \boldsymbol{x}_t \right). \tag{13}$$

Here $\frac{\sqrt{\bar{\alpha}_t}}{1 - \bar{\alpha}_t}$ is the signal-to-noise ratio, recovering the setup of (Sclocchi et al., 2025b). At short times, this quantity diverges and the field pins the symbols to the value associated to the largest entry in $\boldsymbol{x}_t$. At long times, on the other hand, the signal to noise ratio will be close to zero, leading to an input that is uniform over all possible symbols, inducing BP to output the marginal probabilities $\hat{\boldsymbol{x}}_0(\boldsymbol{x}_T) = \mathbb{E}[\boldsymbol{x}_0]$, where the expectation is obtained from the distribution $P_0$ adapted to the one-hot encoding.

### C.3. Denoiser-level experiments

The denoiser-level experiments on our data model (see Fig. 1(b) and Fig. 6) follow a pipeline similar to the denoiser-level experiments on CelebA, with one key difference. In addition to comparing the posterior means of models trained on different datasets (sample-split analysis), we also compare each model prediction against the posterior means of each filtered BP level, $\mathrm{BP}_k$. Concretely, we compute $D_{\mathrm{KL}}(\cdot \mid\mid \hat{\boldsymbol{x}}_0^\theta)$ for multiple choices of $\mathrm{BP}_k$. As before, we apply Eq. (1) to each of the $N_{\mathrm{eval}} = 3000$ clean samples, using $N_{\mathrm{reps}} = 5$ independent noise realizations per sample. The final results are obtained by first averaging over all noisy samples and then averaging over all model pairs formed from 15 models trained on different datasets.

### C.4. Sample-level experiments

We conduct two types of sample-level experiments: (i) experiments based on nearest-neighbor divergence (Fig. 4(a)), and (ii) experiments using the U-turn approach (Fig. 5).

**Nearest-neighbor divergence.** For the nearest-neighbor divergence experiments, we proceed as follows. For a model trained on a dataset of $n$ samples with multiple available training checkpoints, we generate $N_{\mathrm{eval}}$ sequences at each checkpoint. For each generated sequence, we identify its closest counterpart in the training set, defined as the training sequence with the highest overlap—i.e. the highest fraction of agreeing sequence elements, equal to 1 minus the distance normalized by the dimension. This yields an empirical distribution of maximal overlaps with the training data. We then compare it to an analogous distribution computed by, for each of $N_{\mathrm{eval}}$ held-out test sequences drawn from the same underlying data distribution, retrieving the nearest training sequence. Finally, for each checkpoint, we compute the KL divergence between these two distributions. Fig. 4(a) is obtained by setting $N_{\mathrm{eval}} = 50000$ and by averaging the curves obtained for 15 models trained on the same dataset to reduce fluctuations. Figs. 10, 11 (left) represent the evolution of the KL divergence for two distinct models trained on 12000 and 5000 data points respectively.

**U-turn experiments.** For the U-turn approach, we use the following procedure. We first define a grid of diffusion time steps $t_1, \ldots, t_K$ with $t_i \in [1, 500]$ and $t_{i-1} < t_i$. We then select $N_{\mathrm{eval}} = 100$ samples from both the training and test sets. For each sample and each $t_i$, we apply Eq. (1) to generate $N_{\mathrm{reps}} = 1000$ independent noised versions. Given these noised inputs and a model at a chosen checkpoint, we perform U-turns by running the reverse diffusion process starting from the noised samples. For each $t_i$, this yields $N_{\mathrm{eval}} \times N_{\mathrm{reps}}$ generated sequences.

For every generated sequence, we compute its overlap with the corresponding original (clean) sequence. Formally, for a given denoiser $\boldsymbol{f}$ (either the trained model $\hat{\boldsymbol{x}}_0^\theta$ or the BP oracle) and a fixed starting point $\boldsymbol{x}_0$, let $\tilde{\boldsymbol{x}}_0^{(r)}(\boldsymbol{x}_0, t_i; \boldsymbol{f})$ denote the output of the reverse process at $t = 0$ initialized from $\boldsymbol{x}_{t_i}^{(r)} = \sqrt{\overline{\alpha}_{t_i}}\, \boldsymbol{x}_0 + \sqrt{1 - \overline{\alpha}_{t_i}}\, \boldsymbol{\xi}^{(r)}$, $\boldsymbol{\xi}^{(r)} \sim \mathcal{N}(0, \mathbf{I}_d)$, with $r$ indexing independent noise realizations along the forward and reverse paths. The U-turn overlap is

$$\phi_t(\boldsymbol{x}_0; \boldsymbol{f}) = \frac{1}{N_{\text{reps}}} \sum_{r=1}^{N_{\text{reps}}} \frac{\tilde{\boldsymbol{x}}_0^{(r)}(\boldsymbol{x}_0, t; \boldsymbol{f}) \cdot \boldsymbol{x}_0}{d}, \tag{14}$$

with $\phi_t^\theta(\boldsymbol{x}_0) \equiv \phi_t(\boldsymbol{x}_0; \hat{\boldsymbol{x}}_0^\theta)$ and $\phi_t^{\text{BP}}(\boldsymbol{x}_0) \equiv \phi_t(\boldsymbol{x}_0; \hat{\boldsymbol{x}}_0^{\text{BP}})$.

We then average these overlaps over $N_{\text{reps}}$, obtaining an overlap vector of length $N_{\text{eval}}$. We repeat the same procedure using the BP denoiser in place of the model. Finally, we take the element-wise ratio between the model and BP overlap vectors and average this ratio over $N_{\text{eval}}$. We repeat this pipeline for all $t_i$ and across multiple model checkpoints. Figures 10 and 11 (right) report the U-turn results over multiple epochs for two models trained on 12000 and 5000 samples, respectively.

### C.5. Training-free toy denoiser model

As mentioned in Sec. 2.1, an infinitely expressive diffusion model trained from an uninformed (random-sequence) initialization will asymptotically collapse to pure memorization, behaving as the denoiser induced by the empirical data distribution. For discrete-valued sequences, the empirical denoiser can be expressed in a BP / factor-graph formalism by replacing the binary-tree structure of the data model with a single factor

$$\phi(\boldsymbol{s}) = \sum_{\mu=1}^{n} \prod_{i=1}^{N} \delta_{s_i, s_i^\mu}, \tag{15}$$

i.e., a hard constraint that concentrates the (normalized) probability measure on the training set $\{\boldsymbol{s}^\mu\}_{\mu=1}^n$. An architecture-free toy model of the progressive alignment with the empirical denoiser is obtained by considering a *smoothing* of this factor:

$$\phi_\varepsilon(\boldsymbol{s}) = \sum_{\mu=1}^{n} \prod_{i=1}^{N} \left( \delta_{s_i, s_i^\mu} + \mathrm{e}^{-\varepsilon} \left( 1 - \delta_{s_i, s_i^\mu} \right) \right), \tag{16}$$

which allows deviations from the training sequences, with mismatches exponentially suppressed. The parameter $\varepsilon$ controls the sharpness of the distribution and thus the typical distance from the training points at which samples concentrate. After normalization, $\varepsilon = 0$ yields a flat distribution over sequences, while the limit $\varepsilon \to \infty$ recovers the empirical distribution.

What is missing from this simple model is the inductive bias of the trained architecture: the $\varepsilon$-regularization is completely agnostic to the data model, and therefore the interpolation of $P_0$ far from the training points is expected to be poor.

Although one can compute the conditional posterior mean given an observation, as done in Eq. (13), sampling from the smoothed distribution does not require an iterative denoising diffusion mechanism. Indeed, one may simply draw an index $\mu$ uniformly from $\{1, \ldots, n\}$—each mixture component having equal total mass by symmetry—and then sample $\boldsymbol{s}$ from the corresponding product kernel centered at $\boldsymbol{s}^\mu$.

## D. Supplementary figures

### D.1. CelebA images

**Typical sample-split generated images.** In Fig. 8, we show sets of 20 images generated by two models trained according to the sample-split procedure described in B, for the same checkpoints shown in Fig. 1. On the left, close to initializations, the models output different yet already similar noisy samples. By the epoch shown in the center left, both models have improved significantly and get closer together. At the typical minimum of the cosine distance among generated images, essentially all rows are near identical. At the typical minimum of the test loss, some visible discrepancies start arising in some (albeit not all) of the generations.

**Sample-split comparison of models at fixed times.** In Fig. 9, we show additional diffusion times for the comparison of models trained in CelebA, as in Fig. 2. The phenomenology is identical to what is described in Sec. 3. However, for longer

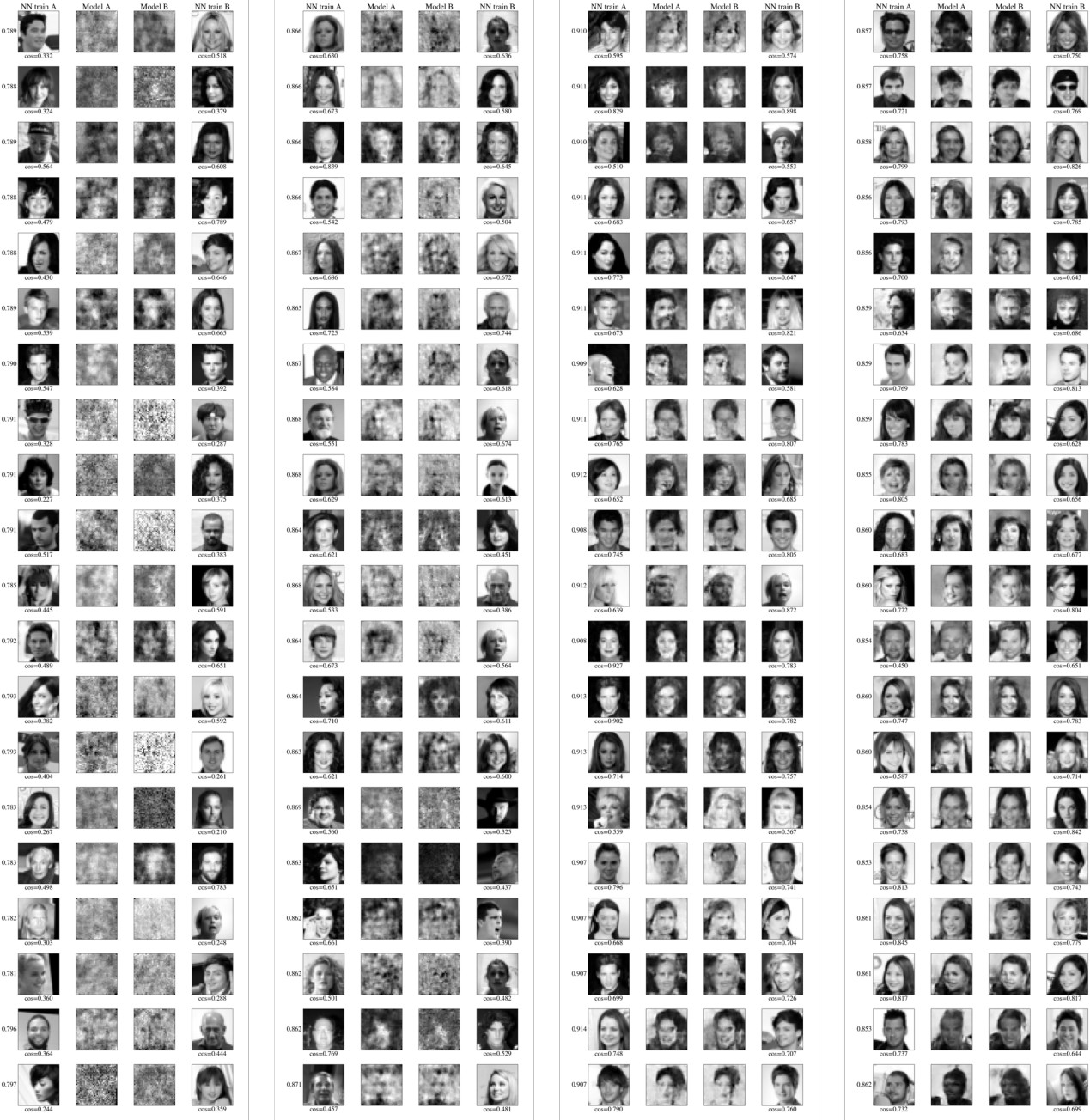

*Figure 8.* Set of the 20 images among 512 with the closest to typical cosine similarities between predicted noises after (from left to right) 111, 213, 620 and 1333 epochs. As in Fig. 1(a) (fourth row from the bottom here), central columns show samples generated by two models given the same noise trajectories, while outer columns show the closest image in their respective training sets. On the left side of each row is displayed the cosine similarity between the generated images, while below outer images is displayed the cosine similarity between the generated images and the associated closest training sample.

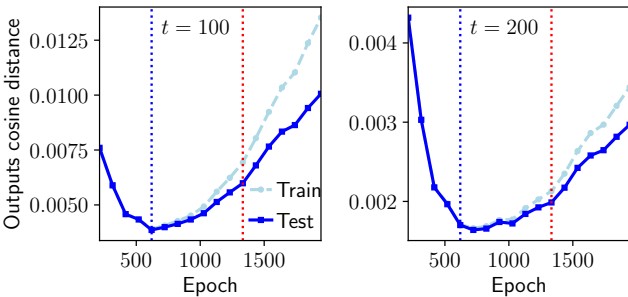

*Figure 9.* Reproduction of the experiment presented in Fig. 2 for two additional fixed diffusion times.

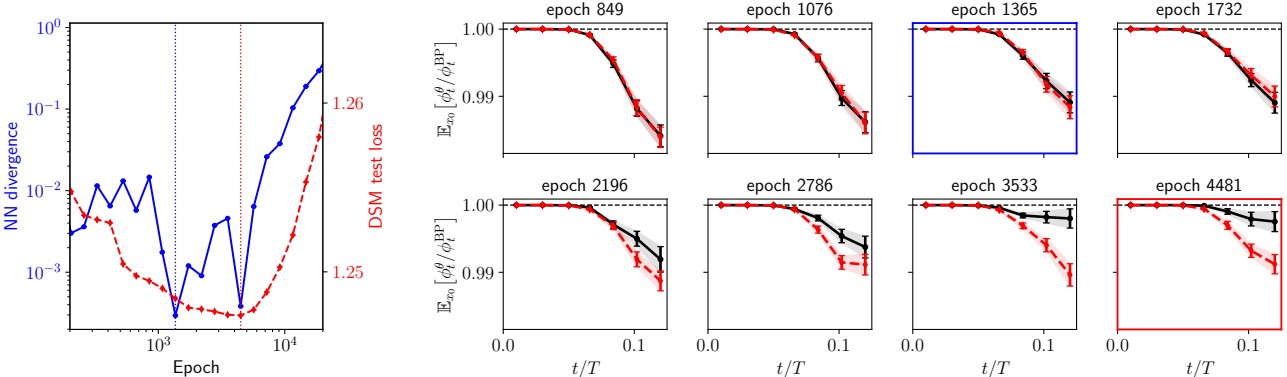

*Figure 10.* Left: NN divergence and DSM test loss versus training epoch for a model trained on $n = 12$k samples of hierarchical data, with respective minima indicated by vertical dotted lines. Right: U-turn overlap ratios across epochs for starting points in the training set of the model (black lines) or from a test set (red dashed lines). Blue/red framed panels highlight the epochs corresponding to the bias metric minimum and test-loss minimum respectively (shown in Fig. 5).

times, we see that the impact of biased generalization is less pronounced in the denoiser's behavior, as expected from the vanishing data-dependence of the long-time regime of diffusion.

### D.2. Hierarchical data model

**Single-instance NN divergence.** In Fig. 10 (left) and Fig. 11 (left), we show two representative seeds of the experiment shown in Fig. 4 (left), where the curves are averaged over 15 independent runs. As can be seen from these curves, the Adam optimization dynamics introduces strong oscillations that significantly affect the nearest-neighbor divergence. While the described phenomenon is robust across runs, the amount of bias recorded at the minimum of the test loss—that can be quantified by looking at the difference between the two values of the blue curve in correspondence of the dashed lines, is training trajectory dependent and can vary from run to run. As also shown in these figures and described just below, other methods of diagnostic such as the U-turn experiments may thus be more robust than this extreme value statistic for single instances.

**Full sequence of U-turns across training.** Figures 10 (right) and 11 (right) report the U-turn results on our data for two models trained on 12,000 and 5,000 training samples, respectively. We find that, at the epoch corresponding to the minimum test loss, the models are closer to BP, even though they exhibit asymmetric behavior between training and test data. In contrast, at earlier epochs the models behave more similarly across the two splits, but this comes at the expense of poorer generalization.

**U-turns on random starting sequences.** We further investigate the behavior of our trained denoisers in the short-time dynamical regime (regime (iii) in Sec. 4.2). In particular, in the main, we described this regime as *trivial*, since all our oracle denoisers and the trained models display a negligible posterior mean divergence $D_{\mathrm{KL}}$ for $t/T \lesssim 0.8$, provided the comparison is performed around valid i.i.d. datapoints $x_0$. Note that the equivalence class could be further extended to the

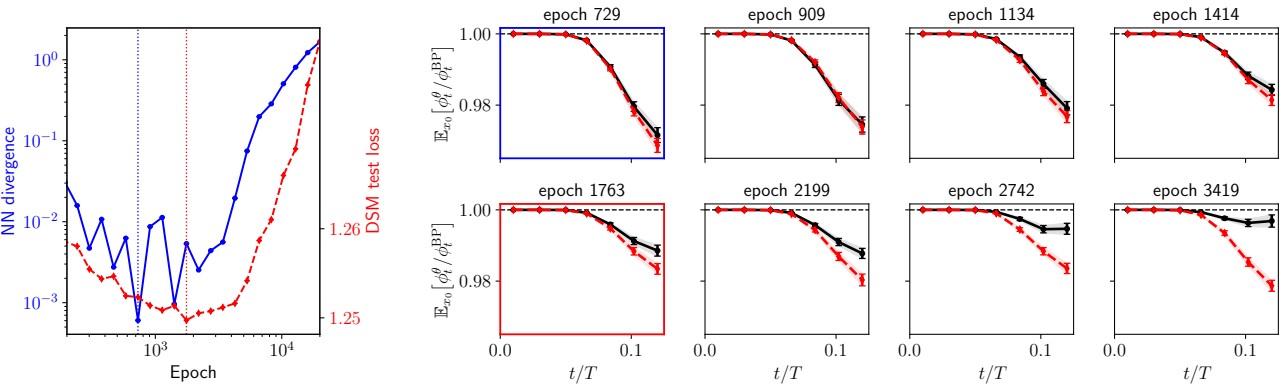

*Figure 11.* Left: NN divergence and DSM test loss versus training epoch for a model trained on $n = 5\text{k}$ samples of hierarchical data, with respective minima indicated by vertical dotted lines. Right: u-turn overlap ratios across epochs for starting points in the training set of the model (black lines) or from a test set (red dashed lines). Blue/red framed panels highlight the epochs corresponding to the bias metric minimum and test-loss minimum respectively.

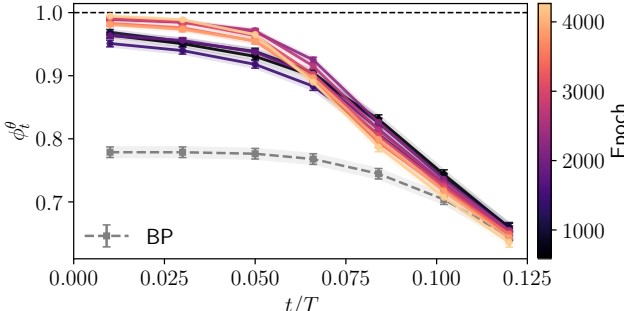

*Figure 12.* U-turn experiment performed by noising sequences whose discrete elements are drawn uniformly at random. The left y-axis shows the averaged overlap of the generated sequences with the original ones.

empirical posterior mean, if the sequences were selected from the training set.

However, we also argued that this similarity would break if we considered a reference $\boldsymbol{x}_0$ that has zero probability in some of the models, and non-zero in others. To investigate this, we follow similar lines to the U-turn experiment described in Sec. 4.3. As an extreme example, we consider reference sequences sampled uniformly at random from $\{1, .., q\}^N$, and their corresponding one-hot encodings $\boldsymbol{x}_0$. What we expect is that a U-turn process starting from noised random sequences could come back to the original point only when it has a non-vanishing probability in the associated posterior measure. Therefore, more selective posterior means (in order: the empirical posterior mean, $\text{BP}_0$, $\text{BP}_1$, ...) should obtain a smaller average overlap $\mathbb{E}[\phi_t(\boldsymbol{x}_0)]$.

In Fig. 12, we show the typical overlap achieved with the BP oracle posterior mean (grey dashed line), and the corresponding overlap obtained with our trained model (different epochs in different colors) from this experiment. The results confirm that the trained model very quickly learns a simpler explanation, here likely rounding up to the closest $\{0, 1\}$ sequence to obtain a one-hot vector, instead of the ground truth. This finding confirms that the trained denoiser is approximating the behavior of BP and the empirical posterior mean "by coincidence" in the trivial phase, since the learned short-time function is compatible with the former denoisers only in proximity to specific points in input space.

