# OpenReview forum: "Biased Generalization in Diffusion Models"
_ICML.cc/2026/Conference — ICML 2026 spotlight_

### Official Review · Reviewer_ZBeu · 2026-03-02

**Soundness:** 3
**Presentation:** 3
**Significance:** 3
**Originality:** 3
**Overall Recommendation:** 5
**Confidence:** 5

**Summary:**

The authors of this manuscript investigate the training dynamics of diffusion models along two axes: generalization and memorization. In contrast to other works, they argue that these two phenomena are not mutually exclusive, but orthogonal axes describing the behavior of diffusion models. They demonstrate that diffusion models trained on disjoint datasets initially implement an increasingly similar mapping. They then show that the dissimilarity between the two models begins to increase before the test loss has reached its minimum. They call this phase of training biased generalization, which they reproduce using a discrete diffusion model trained on a context-free grammar. For those models trained on the context free-grammar, they show that these models initially are closer to optimal denoisers, taking only coarser features into account than those using also finer features.

**Compliance With Llm Reviewing Policy:**

Affirmed.

**Final Justification:**

The authors have answered all my questions. I maintain my "accept" score.

**Key Questions For Authors:**

Regarding 3):

Can the equivalent of point (2) also be shown for non-synthetic data, e.g. the CelebA dataset studied by the authors? For example, using projections onto leading/subleading PCA directions as coarser/finer-level features? This could support the similarity between the author's heuristic model and effects on real data, i.e. support their mechanistic explanation of the effect.

Regarding 4):

Can it be verified that memorization sets on (also in the CelebA data), when the test loss increases?
Or do the authors equate memorization with overfitting? In this case, which training loss do these models achieve? Could one not argue that overfitting occurs when training and test loss begin to deviate? Does this occur at the same point at which the test loss is minimal?

Minor points:

5) The training dynamics of generative and in particular, diffusion models have been studied in the linear/Gaussian regime by refs. [1-3] (see below). All three references show that directions corresponding to leading eigenvalues of the data covariance matrix are learned earlier, while subleading eigendirections are learned later. Moreover, in images, the leading eigenspace is known to be responsible for coarser, rather than finer features. Could the early alignment of the leading subspace be responsible for the existence of the biased generalization phase?

6) Can the authors state the exact loss functions that they use for their experiments on both datasets? Are they the same for both experiments? In the practice of training diffusion models, the loss function can be the mean squared error of the model predicting the additive noise, the original data, or a linear combination of the two. The use of these different objectives can lead to different values for the test losses.

[1] A theoretical framework for overfitting in energy-based modeling, Catania et al.

[2] Generalization Dynamics of Linear Diffusion Models, Merger et al.

[3] An Analytical Theory of Spectral Bias in the Learning Dynamics of Diffusion Models, Wang et al.

**Limitations:**

yes

**Strengths And Weaknesses:**

Strengths
1) The manuscript makes an interesting point: The existence the training phase they call biased generalization.

2) They show that in this training phase the model trained on a context free-grammar learns coarser features but not finer features.

Weaknesses

3) Point (2) is only shown to hold on synthetic data.

4) The manuscript uses an increasing test loss as a proxy measure for memorization, but do not demonstrate that it is a good proxy measure. This is an important check to validate the soundness of the central result.

---

> ### Author Rebuttal · Authors · 2026-03-30
>
> We thank the reviewer for their interest in our work and their constructive feedback.
>
> ---
>
> *Regarding 3)* question & *Minor 5*. We warmly thank the reviewer for motivating us to explore the connection with the deep-linear theory and for the pointers to Catania et al., Merger et al., and Wang et al., which we will reference in the revised version as complementary references for sequential learning. In particular, we followed their suggestion to consider "filtering" protocols also for the CelebA data, to build surrogates for the BPk oracles used in the synthetic setting.
>
> We first tested the reviewer’s exact suggestion of projecting the data onto the leading PCA components (retaining k = 32, 128, 512, 1024 modes). Our preliminary experiments seem to show that for a spatial U-Net, PCA is not the most appropriate basis to separate ”coarse” from ”fine” features: since the top 32 components already capture ∼90% of the dataset’s global variance, the local convolutional feature maps remain largely unperturbed, and cross-level model similarities remain pinned at \sim0.99 throughout training (this might change with different numbers of retained modes).
>
> Instead, to better match the inductive bias of convolutional architectures, we also considered a 3-level Haar Wavelet decomposition. By zeroing out the finest high-frequency detail scales, we created 4 levels of structural complexity (from L0, keeping only the blurry 4 × 4 base approximation, to L3, the unfiltered original images). We trained independent models on these filtered datasets and froze
> them at their respective minimum test loss checkpoints. We then tracked the score predictions of a standard unfiltered model during training against these frozen targets. Using MSE, we observed that the peak of similarity is found earlier for the simpler approximations and later for the more complete ones. Moreover, there are some indications—to be corroborated with more statistics—that the models trained on more complex data develop a higher degree of training data bias (we are deducing this from the behavior of the sample-split divergence on training/test data).
>
> However, we believe that an explanation based solely on first and second moments (as in the deep-linear covariance-spectrum-based picture) has a fundamental limitation in capturing the higher-order correlations entailed in a hierarchical structure. While the modes are learned on different timescales, they are learned in parallel. To observe something more reminiscent of our BPk comparisons with the synthetic data, we are planning to explore an additional non-linear filtering protocol, based on training convolutional auto-encoders with different numbers of convolutional blocks in the decoder segment. The idea is to follow an approach similar to Refinetti et al. (2023) and Gerace et al (2024) and obtain clones of the CelebA dataset with a controllable amount of features.
>
> ---
>
> *Regarding 4)* question. Recent works, Bonnaire et al. (2025) and Favero et al. (2025b) in particular, have established theoretically and experimentally that the typical metric for detecting memorization in diffusion models (a sufficiently large ratio of the overlap to the closest and second closest samples in the training set) starts increasing in correspondence with the increase of the test loss. Indeed, in this work, we argue that the bias toward training data can start developing much earlier, when training and test loss start to separate—see loss decomposition in Sec. 4.3. Importantly, this happens strictly before the minimum of the test loss. The gap between these timings can shrink when the model has the correct inductive bias and when there is more data available for training, since more and more features will be absorbed in an unbiased fashion, but this does not prevent bias from affecting the generation—especially if guidance heuristics are employed.
>
> ---
>
> *Minor 6*. We thank the reviewer for raising this point. In the revised version, we will explicitly state the learning objectives. In fact, in our experiments on the UNet and CelebA data, the models predict the additive noise (same diffusion backbone as in Bonnaire et al (2026)); in the experiments on the transformer model and discrete hierarchical data, instead, we follow Li et al. (2022) and train on the posterior mean prediction—this also allows a more natural parallel with the BP oracle and its estimation of the posterior marginals.
>
> **Additional References**
>
> Rende et al. (2024): Riccardo Rende, Federica Gerace, Alessandro Laio, and Sebastian Goldt, "A distributional simplicity bias
> in the learning dynamics of transformers", Neurips 2024
>
> Gerace et al. (2024). Federica Gerace, Diego Doimo, Stefano Sarao Mannelli, Luca Saglietti, Alessandro Laio, "Optimal transfer protocol by incremental layer defrosting", TMLR 2024.

---

> > ### Author Rebuttal · Reviewer_ZBeu · 2026-04-01
> >
> > I thank the authors for their response and I look forward to seeing the results of their additional numerical experiments.
> >
> > I would like ask a final question about point 4). While I agree that the works by Bonnaire et al. (2025) and Favero et al. (2025b) show that memorization typically increases after the minimum in the test loss, exact memorization of training samples is perhaps the worst instance of overfitting. The observation of the authors, that overfitting in the form of biased generation, takes on before memorization, is hence entirely plausible. Is there a difference between using the difference between test and training loss, or the new measure introduced by the authors, regarding the onset of overfitting?

---

> > > ### Author Response · Authors · 2026-04-02
> > >
> > > We thank the reviewer for this insightful follow-up. The train/test loss difference is indeed a natural candidate for detecting overfitting, but in practice it suffers from a fundamental ambiguity at finite dataset sizes: the ordering between test and train losses on specific instances is not guaranteed due to sampling fluctuations, making it difficult to systematically attribute a measured gap to genuine bias rather than to statistical noise in the composition of the two sets. Our loss decomposition in Eq. (4) resolves this ambiguity: the excess data-dependent term isolates the component of the loss that is driven specifically by the pull of the empirical distribution, and cannot be explained by an improvement in inference quality. That said, we agree that if a sufficiently large and persistent train/test gap is observed, bias is very likely the driving factor---but the threshold for "sufficiently large" is itself subject to the finite-size uncertainty we describe.

---

### Official Review · Reviewer_UFCg · 2026-03-08

**Soundness:** 3
**Presentation:** 3
**Significance:** 2
**Originality:** 2
**Overall Recommendation:** 4
**Confidence:** 4

**Summary:**

This paper introduces a counter-intuitive phenomenon in the study of generalization in diffusion models: "before the test loss has reached its minimum, the model already tends to generate patterns similar to certain training data."  This paper introduces a counter-intuitive phenomenon in the study of generalization in diffusion models: "before the test loss has reached its minimum, the model already tends to generate patterns similar to certain training data."  Based on this, they develop a more rigorous evaluation framework.

**Compliance With Llm Reviewing Policy:**

Affirmed.

**Final Justification:**

In my discussion with the authors, most of my concerns have been resolved, particularly the authors' commitment to discussing the related work. Furthermore, I believe that this work requires a more in-depth discussion of experimental analyses (e.g., on SDXL/Flux) in the future. I also respectfully argue that theoretical work needs to be validated through applications, which would help ensure the validity of the theory. Based on this, I will maintain a weak accept.

**Key Questions For Authors:**

Please see **Strengths And Weaknesses**.

**Limitations:**

Yes

**Strengths And Weaknesses:**

**Strength**:

1.The paper presents a thought-provoking perspective that challenges the conventional point[1] regarding the alignment between generalization and the minimum of the test loss. This counter-intuitive hypothesis is plausible, and the authors support it with elegantly designed experiments that validate their claims.

2. The authors provide an detailed analysis of the training dynamics, characterizing the temporal evolution of the model’s behavior. By mapping how bias gradually accumulates over time, the paper offers valuable insights that will undoubtedly serve as a foundation for future research in this direction.

3.  The manuscript is well-presented. For instance, the sub-figures in Figure 1 allows for a clear grasp of the paper's contributions. While the captions are quite dense, they do not detract from the clarity of the presentation.

---

**Weakness**:

1.The paper lacks a dedicated "Related Work" section, which makes it difficult to objectively assess the novelty and specific contributions of this work within the broader literature.

2.The empirical evidence is primarily derived from CelebA and synthetic data. While these suffice for mechanism validation, it remains unclear whether these findings extrapolate to large-scale generative models (e.g., SDXL or Flux). Specifically, it is questionable whether the distinction between "coarse" and "fine-grained" hierarchical features remains as clearly defined in models with massive parameter counts and complex data distributions.

3.While the analysis and diagnostic frameworks are exemplary, the paper stops short of proposing concrete strategies to address the identified bias (a point acknowledged by the authors in the limitations). I believe that for interpretability-focused work to be fully validated, it should be complemented by practical applications or interventions to demonstrate its utility in real-world scenarios.

---

[1] Favero, Alessandro, Antonio Sclocchi, and Matthieu Wyart. "Bigger Isn't Always Memorizing: Early Stopping Overparameterized Diffusion Models." arXiv preprint arXiv:2505.16959 (2025).

---

> ### Author Rebuttal · Authors · 2026-03-30
>
> We thank the reviewer for their positive assessment and detailed feedback.
>
> ---
>
> A dedicated related work section would indeed improve the paper and, we believe, also address the reviewer’s mild concerns about originality and significance. We had included such a section in an earlier draft but removed it to satisfy the 8-page limit; we plan to restore it in the camera-ready version. Its main structure would be:
>
> (i) *Memorization vs. overfitting in generative models*. Van den Burg & Williams (2021) show that VAEs can remain sensitive to their training set without replicating examples, a conceptual precursor to our findings, though in a different model class and without controlled analysis of the dynamics. Yoon et al. (2023) argue that memorization and generalization are mutually exclusive in diffusion models; our results directly challenge this. Chae et al. (2025), which we became aware of only after submission, study benign overfitting in overparameterized generative models. Our results suggest caution with this framing: any train/test discrepancy implies data-dependent behavior, which in the generative setting already reflects bias toward training samples, even when aggregate metrics improve.
>
> (ii) *Generalization-to-memorization transition in diffusion*. Bonnaire et al. (2025), Favero et al. (2025a,b), George et al. (2025), and Li et al. (2023) study the transition from generalization to memorization and place its onset at or after the test-loss minimum. Our contribution is to show that data-dependent bias appears earlier.
>
> (iii) *Memorization detection and privacy*. Carlini et al. (2023), Somepalli et al. (2023), Ross et al. (2025), and Gu et al. (2025) develop tools to detect or characterize memorization. Our diagnostics complement these by detecting softer bias that precedes overt memorization.
>
> (iv) *Test loss as a generalization metric*. Independent of and after our submission, Mendes et al. (2026) show in a solvable autoencoder model that decreasing test loss can coexist with degrading representation quality, closely echoing our findings. More fundamentally, Biroli & Mézard (2026) show in an exactly solvable high-dimensional KDE problem that even optimizing KL divergence does not preclude bias: beyond a phase transition, the optimal estimator becomes strongly data-dependent. As discussed in Sec. 5.2, this supports the broader relevance of our conclusions.
>
> ---
>
> Regarding scale, we agree that extending the analysis to models such as SDXL or Flux is important. However, these are conditional models that rely on guidance mechanisms (classifier-free guidance/guidance distillation), so understanding how guidance interacts with biased generalization is likely a prerequisite; we highlight this as an open direction in Sec. 7. More broadly, our goal here was to establish the existence of the phenomenon in a setting where exact scores, ground-truth statistics, and exhaustive split analyses are available. We view this as a necessary first step before moving to state-of-the-art systems. While feature-learning dynamics will of course be less schematic in high-dimensional data, our controlled setting shows that biased generalization is a concrete possibility and provides diagnostics that make it visible. We also note that Biroli & Mézard (2026) show that minimizing KL divergence does not, in general, prevent training-data bias even in infinite dimensions, suggesting that the mechanism is not a finite-dimensional artifact.
>
> ---
>
> On the final point, we agree that mitigation is an important direction and explicitly list it as a limitation in Sec. 6. That said, we respectfully disagree that diagnostic work must already include a mitigation strategy to be useful. In our case, the existence of biased generalization had not previously been established, and identifying a failure mode that escapes standard evaluation metrics is already practically relevant. In particular, for limited-data regimes, where the effect is strongest, the message that early stopping at the test-loss minimum may be insufficient for privacy-sensitive applications is already actionable. We therefore hope this work helps motivate further research on mitigation and on safer generative modeling more broadly.
>
> **Additional References**
>
> Chae et al. (2025): Jiseok Chae and Kyuwon Kim and Donghwan Kim, "Rethinking Memorization–Generalization Trade-Off in Generative Models", High-dimensional Learning Dynamics 2025, https://openreview.net/forum?id=JIMZsqE8bA
>
> Mendes et al. (2026): Mendes, V. C., Bardone, L., Koller, C., Moreira, J. M., Erba, V., Troiani, E., & Zdeborová, L. (2026). "A solvable high-dimensional model where nonlinear autoencoders learn structure invisible to PCA while test loss misaligns with generalization." arXiv preprint arXiv:2602.10680.

---

> > ### Author Rebuttal · Reviewer_UFCg · 2026-04-01
> >
> > Thank you for your response. My concerns have been addressed. I would kindly suggest including a discussion of related work in the appendix.
> >
> > As I have already given a positive score, I will not adjust the rating.

---

### Official Review · Reviewer_KLDm · 2026-03-12

**Soundness:** 3
**Presentation:** 2
**Significance:** 4
**Originality:** 4
**Overall Recommendation:** 5
**Confidence:** 4

**Summary:**

The authors investigate the generalization behaviour of diffusion models. Specifically, they challenge the commonly held belief that memorization in diffusion models only occurs after the model as achieved minimum test loss. In two domains, the authors identify a region of memorization which begins prior to test loss minimization. By analyzing model behaviour on disjoint training subsets, they find that within this region, model performance differs based on their training set, but not in a way which leads to adverse test set performance. The authors investigate this phenomenon in both image diffusion models and discrete diffusion models. In both cases, then analyze bias in both the generated samples and the denoising functions, with observations that support their hypothesis.

**Compliance With Llm Reviewing Policy:**

Affirmed.

**Final Justification:**

Overall, I think this is a good paper which advances our understanding of diffusion model generalization. I think the experiments are well formulated and executed. While I had reservations in my review about the technical clarity of the paper, these reservations have been addressed by the authors who have committed to clarifying details in the camera ready revision.

On this basis, I think this paper should clearly be accepted to ICML.

**Key Questions For Authors:**

I've selected a couple of points from the weaknesses highlighted above that I believe should be addressed first by the author

## Q1
How do you define bias formally at the distributional, and estimator level?

## Q2
Is precisise bias estimation possible in all settings? What challenges present themseleves?

## Q3
How do you define nearest neighbour divergence?

**Limitations:**

yes

**Strengths And Weaknesses:**

Overall, I really enjoyed this paper. I think the findings are significant, and run contrary to a commonly held beliefs in both the community of practitioners who train and deploy diffusion models, and the community which is investigating their generalization behaviour. I thought the experiments were thoughtful and supported the claims of the paper well. My main complaint of the paper is the lack of precision in language throughout. I believe these concerns can be addressed fairly easily, leading to my support for the acceptance of this work.

Below, I have grouped my detailed feedback into strengths and weaknesses in each of the evaluated dimensions:

## Soundness

**Strengths**

 - As I stated in my high-level comment, I think the design of the empirical experiments are a real strength of this work. They provide clear evidence supporting the hypothesis of the paper. In particular, I think figure 5 is a compelling result regarding how bias is introduced at the test loss minimum
- I appreciated the authors' commitment to using error bars in many of their results

**Weaknesses**

  - For a paper which has "Biased" as the first word of the title, I found the authors' definition of bias in 2.2 rather flimsy. I would expect bias to be formally defined using equations and not simply through half a page of prose. I understand that in some contexts the true data distribution or score is unknown making explicit bias calculations intractable. However, this is not always the case (as shown in the hierarchical setting). I would much prefer the authors rooted their discussion in a definition and then elaborated about how to approximate the quantity when the true data distribution is unknown.
- The definition of "nearest neighbour divergence" (L149) is unclear to me. It seems like the authors are comparing a divergence between the distribution of distances to the nearest training set image over a distribution of training images and sampled images. However, it is unclear to me how those distributions are modelled? Is it a kernel density estimate? Again, more rigour here would benefit the impact of the paper.
- Nit: In the hierarchical setting, section 4.3 you discuss KL divergences between posterior means. While I understand that since x can be viewed as N q-way categorical distributions, we can view the posterior mean as a similar set of N probability distributions, I think the language used is imprecise - KL divergences are taken between distributions, not means. Furthermore, It is not clear what factorization of x is being employed. Are the authors assuming independent distributions per symbol? Ie $p(x_0 | x_t) = \prod_{i=1}^N p(x_{0,i} | x_t)$?.

## Presentation

**Strengths**

 - Overall, I thought the narrative of the paper was strong
 - I found the writing quality to be high

**Weaknesses**

 - The authors state on L133 that they will interchangeably refer to equation (2) loosely as a score function. To be clear, this is not a score function. It is relatively easy to be precise when choosing which quantity to talk about (score, velocity, posterior mean/denoiser). I think blurring the line between these quantities adds confusion. For example, when reporting DSM loss, are you reporting an L2 error between x predictions or score predictions? The same question could be asked of the results of figure 2. I understand that the authors wish to distinguish between bias at the final-sample level, and at the level of score-level, where score-level is evaluation of a single network outptut. Perhaps alternate language such as "estimator-level" or "function-level" might help to avoid overloading score in the paper.
- I think the one section which seems somewhat out of place in the narrative of the work is section 4.2 I am not sure if this section supports the hypothesis around train-set specific bias. In my view it mostly just confirms prior findings regarding the three dominant regimes of the diffusion process.
- It was unclear to me how to compute an overlap based on the real-valued sampled data. Do the authors argmax their sampled data to get a discrete sample from their model? This is another case where a simple equation would really improve clarity.
- In the same vein, $\phi_t(x_0)$ would also benefit from a formal definition.

- Nit: L82 paragraph is broken unevenly across the column break
- Nit: The authors cite Kamb & Ganguli 2024 when discussing favorable local inductive biases, but do not cite the concurrent work which also identified this bias - Niedoba et al. "Towards a Mechanistic Explanation of Diffusion Generalization" 2024.
- Nit: L221 I think the distribution should not be referred to as marginal - is it not a posterior distribution given the observation $x_t$?

## Signficance

**Strengths**

 - Intellectually, I think this work is significant in the literature, as it challenges a commonly held notion that memorization and generalization are purely dependent. I think this contribution is valuable to the community investigating diffusion generalization
- Practically, I think this work is significant for users of diffusion models, as it provides evidence that minimum test-loss may not be a good metric for model seleciton.

**Weaknesses**

 - While the authors analyze both images and hierarchical discrete data, they spend the majority of the paper analyzing the discrete data regime. While understandable, this is a fairly niche data domain. The results on this domain support their hypothesis but may not be as directly applicable as those on the image domain.

## Originality

**Strengths**

  - I think this is a novel direction of inquiry inside the generalization field

---

> ### Author Rebuttal · Authors · 2026-03-30
>
> We thank the reviewer very much for their very detailed feedback, as well as for their overall very positive assessment of our work. Let us first answer their specific questions:
>
> **Q1.** On a formal definition of bias, we propose the following (along the lines of Carlini et al. (2023)): given a metric $d$, a radius $\delta$, and a tolerance $\lambda$, we say that a generative model $\\hat{P}$ is $(d,\\delta, \\lambda)$-biased towards its training data $\\{\\boldsymbol{x}^\\mu\\}$ relative to the true data model $P$ if there exists $\mu$ such that $\Pr_{x\sim\hat{P}}[d(\boldsymbol{x}, \boldsymbol{x}^\mu) < \delta] - \Pr_{x\sim P}[d(\boldsymbol{x}, \boldsymbol{x}^\mu) < \delta] > \lambda$. Alternatively, the definition could be made at the estimator level, comparing the learned denoiser to either the ground-truth score or to a denoiser trained on independent data. However, both formulations suffer from the same intrinsic difficulty: they displace the core issue onto the choice of similarity/distance metric and parameters, which in practice is highly context-dependent—from a differential privacy perspective, any nonzero mutual information between the model and training data constitutes bias, whereas from a generative fidelity standpoint, a much coarser threshold is acceptable. If the reviewer sees fit, we are happy to include such definitions in Sec. 2.2 and relate them to our metrics, while highlighting that the choice of these parameters is application-dependent and that we choose to focus on characterizing bias and demonstrating that it emerges before traditional signs of overfitting.
>
> ---
>
> **Q2.** Precise bias estimation is in principle possible in all settings through the sample-split methodology, which requires only the ability to train models on independent data subsets. However, several practical challenges arise. First, averaging score divergence over all diffusion times can wash out the signal, as bias manifests most clearly at intermediate times where denoising is input-dependent—identifying this informative range is in fact the role of Sec. 4.2, which we believe justifies its inclusion despite the reviewer's concern. In practice, one should explore ranges of fixed diffusion times rather than relying on time-averaged metrics. Second, the method requires sufficient data to construct meaningful disjoint splits. Third, adaptive optimizers can induce significant fluctuations in bias metrics (see Figs. 9–10 for NN divergences where the effect is more dramatic), so reliable estimation along training may require averaging over multiple training runs.
>
> ---
>
> **Q3.** We agree that the definition was introduced too quickly and will expand on it in the revised version. To summarize: for each generated sample, we record the distance to its nearest neighbor in the training set, building an empirical distribution which we compare via KL divergence to the analogous distribution for i.i.d. ground-truth samples. We use this metric only for discrete data, where overlap and Euclidean distance are in fact strictly equivalent, and no further modeling is needed since the overlap can take a finite number of possible values. In principle, it extends to $\\mathbb{R}^d$, but would then require distribution estimation over a continuous support, introducing the modeling choices the reviewer rightly flags; for images, we therefore use the sample-split methodology with cosine similarity instead. We clarify that overlap is simply $\\boldsymbol{x} \\cdot \\boldsymbol{y} / \sqrt{d} $, i.e. cosine similarity for vectors on a sphere of radius $\\sqrt{d}$.
>
> ---
>
> Let us now address the remaining points. We agree that using "score" interchangeably with the posterior mean is imprecise and will adopt more careful terminology in the revised version, also specifying for each figure what quantity is being compared. We will also provide a formal definition of $\\phi_t(\\boldsymbol{x}_0)$.
>
> ---
>
> Regarding the KL divergence between posterior means and the factorization question: there is no assumption of independence across symbols. Belief propagation expresses the exact joint distribution between the variables in terms of cavity messages, which in turn allow one to express the per-variable marginals. Given that we use a one-hot encoding for the diffusion setting, the posterior mean $\\hat{\\boldsymbol{x}}_0(\\boldsymbol{x}_t) = \\mathbb{E}[\\boldsymbol{x}_0 \\mid \\boldsymbol{x}_t]$ yields a set of $N$ marginal probability vectors ($q$-dimensional). Thus, we compute KL-divergences at the level of the per-symbol marginals, and then average over the $N$ positions.
>
> ---
>
> We thank the reviewer for pointing out Niedoba et al. (2024) and will include it in the revision. Finally, regarding the focus on the discrete regime: we are expanding our exploration of the impact of sequential learning on real continuous images. We refer the reviewer to our response to Reviewer ZBeu for details.

---

> > ### Author Rebuttal · Reviewer_KLDm · 2026-04-02
> >
> > I think the authors' have mostly answered my questions. I think more clarity in the definition of bias would improve the paper and I suggest the authors introduce this into the camera ready draft. Similarly, I appreciate the offer to clarify the concepts of NN divergence and $\phi(x_0)$.
> >
> > As a very minor pushback - I think your reply to my point about the KL is inconsistent. You mention that you do not make a assumption of independence across symbols, but you also state that you are taking a average over the N q-way categorical distribution. Is this not a independence assumption? Obviously modelling the joint with your model is intractable, so you've decided to factorize it by symbol.
> >
> > Regardless, I think this is a good paper and that it should be accepted. I'll be maintaining my score.

---

> > > ### Author Response · Authors · 2026-04-03
> > >
> > > We thank the reviewer for the positive assessment and for pointing out that our previous reply on the KL divergence was not sufficiently clear. We agree that the quantity we report is based on a factorized representation across sites, and we should have distinguished this more explicitly from an independence assumption in the underlying inference.
> > >
> > > In the DDPM setting, the denoiser is trained to predict $\hat{\boldsymbol{x}}_0(\boldsymbol{x}_t)$, which in the discrete case can be interpreted as an $N$-dimensional collection of $q$-way conditional marginals (one categorical distribution per site). Our reported KL is computed by averaging the $q$-way KL divergences between these predicted and target one-site marginals. Equivalently, this can be viewed as the KL between the corresponding products of marginals (i.e., a factorized representation of the posterior).
> > >
> > > However, this factorization enters only at the level of the output representation / evaluation metric, not at the level of the inference procedure. In the $\mathrm{BP}_0$ formalism, the exact one-site conditional marginals are obtained from fixed-point cavity messages on the factor graph. On a tree, this is exact inference, and these same messages also contain the information needed to recover higher-order statistics. Thus, $\mathrm{BP}_0$ does not rely on an a priori independence assumption over symbols; the factorized object appears only because the DDPM observable of interest is the collection of one-site marginals.
> > >
> > > Likewise, the transformer denoiser is trained to approximate these one-site marginals, not the full joint conditional posterior, which would indeed be intractable to represent or compare directly. Our use of the $q$-way KL divergence rather than, e.g., an MSE is simply a natural choice for categorical / one-hot outputs. We agree that this point deserves clearer wording in the paper, and we will revise the camera-ready version to make explicit that the reported KL concerns a factorized marginal representation, rather than an independence assumption in the model or oracle.

---

### Official Review · Reviewer_d1Mq · 2026-03-16

**Soundness:** 3
**Presentation:** 4
**Significance:** 3
**Originality:** 4
**Overall Recommendation:** 5
**Confidence:** 4

**Summary:**

The paper concerns generalization behavior in diffusion models and how it relates to dependence on training data. The authors introduce the concept of biased generalization, a regime in which diffusion models continue to improve their denoising score matching (DSM) test loss while their generated samples become increasingly biased toward training examples.

Through experiments on real images and a controlled hierarchical data model with known ground-truth scores, the authors show that this bias emerges before the test loss reaches its minimum. They attribute the effect to sequential feature learning, where models learn coarse structures first and later approximate finer features in a data-dependent manner. The findings suggest that generalization and memorization can coexist, and that early stopping based on test loss may not prevent training-data bias.

**Compliance With Llm Reviewing Policy:**

Affirmed.

**Final Justification:**

This is a good paper. I recommend acceptance.

**Key Questions For Authors:**

Please address the weaknesses and the following questions/suggestions
- Experiments clarifying the roles of the following factors
    - Scale
    - Architectures
    - Training objectives
    - Guidance methods

**Limitations:**

yes

**Strengths And Weaknesses:**

Strengths:
- The concept of biased generalization is novel and interesting. The work highlights that distributional generalization and dependence on training samples can evolve along separate axes, challenging the common assumption that memorization only emerges after generalization breaks down.
- The evaluation metrics and experiments are well designed, covering a wide range of settings. They provide a more mechanistic understanding of the phenomenon.

Weaknesses:
- Large scale experiments are lacking. Large models trained on large datasets have different generalization behavior (e.g. the emergent phenomenon). It remains unclear whether the same phenomenon appears in large modern diffusion models trained on high-resolution datasets.
- Mitigation strategies were not dicussed.

---

> ### Author Rebuttal · Authors · 2026-03-30
>
> We thank the reviewer for their interest in our work and their overwhelmingly positive feedback. Regarding the four factors mentioned in their questions, we believe that the first three—architectures, training objectives, and scale—are in fact connected through the statistical mechanism we describe in Sec. 5. We address them jointly below, before turning to guidance separately.
>
> ---
>
> The evidence we provide for biased generalization currently spans three qualitatively different settings: (i) a U-Net denoiser trained with an L2 noise-prediction objective on CelebA images, (ii) a transformer denoiser trained with a cross-entropy loss on discrete hierarchical sequences, and (iii) a training-free generative model generalizing Kernel Density Estimation to discrete data.
> These three cases differ in architecture, training objective, and, in the case of (iii), in the complete absence of both. The consistency of the phenomenon across these settings reflects the fact that biased generalization originates from a statistical mechanism rather than from the specifics of any given setup. As discussed in Sec. 5, whenever a model attempts to resolve correlations at scales
> where the available finite data is insufficient to determine a unique, data-independent solution, its outputs necessarily become data-dependent while potentially still improving on distributional metrics. This is the core of the effect.
> The role of architecture and training objective then primarily modulates when and how severely the effect manifests. In deep networks, the sequential nature of feature learning—which is by now widely supported in the literature (Refinetti et al., 2023; Bardone et al., 2025)—means that coarse, data-independent features are learned first, effectively delaying the onset of biased generalization to later stages of training where finer features begin to be resolved. Models that lack such sequential
> learning, like the KDE of Sec. 5.2, have no mechanism to postpone the effect and accordingly display it much earlier and more prominently, see Fig. 1(c). This suggests that favorable inductive biases do not eliminate biased generalization but compress and delay its onset; a prediction that could be tested systematically across architectures. However, such a thorough and computationally
> demanding characterization, in our opinion, lies outside the scope of this paper.
>
> Nonetheless, in the camera-ready version, we propose to expand the discussion in Sec. 5.2 to explicitly contrast the severity of biased generalization in the training-free model, which lacks any architectural bias toward sequential learning, with the trained neural network settings, where the effect is significantly more contained. We would additionally strengthen the conclusion by highlighting the differences between the CelebA and synthetic data setups to further underline the robustness of the phenomenon across modalities.
>
> Regarding scale specifically, we would highlight that the relevant question is whether, for a given data distribution and model, there exists a regime of training where some structural scales can be resolved and others cannot. For any structured distribution with multi-scale features, we expect such a regime to exist for appropriate values of the data dimension $d$ and training set size $n$. The precise scaling of these boundaries with $d$ and $n$ is likely to be data-model-dependent and is
> an important open question that we will more explicitly flag in our limitations. One important piece of information comes from the asymptotic analysis of Biroli & Mézard (2025): they show that minimizing the KL divergence between the model and the data distribution does not preclude bias toward training samples, and this effect is not a finite-size artifact; it persists even in the limit of large dimensions.
> We do not know of a scenario where a qualitative change at a larger scale, i.e. a sudden emergent phenomenon, could lead to different behavior.
>
> ---
>
> Regarding guidance: we fully agree that this is a compelling direction for future work, as discussed in Sec. 7. Classifier-free guidance steers generation toward fine-grained, class-specific features, which are precisely the features most likely to be resolved in a data-dependent manner during biased generalization. We expect guidance to amplify the effect, but confirming this requires a dedicated investigation that is left for future work.
>
> ---
>
> We believe the primary contribution of this work is to identify biased generalization as a distinct phase of training and to provide a mechanistic understanding of its origin, grounded in the interplay between finite data and sequential feature learning. We hope this framework will trigger the systematic investigation of the quantitative role of architecture, scale, and training procedures in
> future research.

---

> > ### Author Rebuttal · Reviewer_d1Mq · 2026-04-03
> >
> > Thank you for your rebuttal. Please include the suggested improvement in the final version.

---

### Decision · Program_Chairs · 2026-04-30

**Decision:**

Accept (spotlight)

**Comment:**

This paper argues that memorization in diffusion models starts to emerge before the test loss achieves its minimal value during training. The authors presented experiments which all reviewers agreed strongly supports this hypothesis. Despite the authors not presenting a mitigation strategy, all reviewers have strongly positive views of the paper. I thus recommend acceptance.